Review

# Cellular stress and epigenetic regulation in adult stem cells

Joey Llewellyn*, Rithvik Baratam*, Luka Culig, Isabel Beerman

**Stem cells are a unique class of cells that possess the ability to differentiate and self-renew, enabling them to repair and replenish tissues. To protect and maintain the potential of stem cells, the cells and the environment surrounding these cells (stem cell niche) are highly responsive and tightly regulated. However, various stresses can affect the stem cells and their niches. These stresses are both systemic and cellular and can arise from intrinsic or extrinsic factors which would have strong implications on overall aging and certain disease states. Therefore, understanding the breadth of drivers, namely epigenetic alterations, involved in cellular stress is important for the development of interventions aimed at maintaining healthy stem cells and tissue homeostasis. In this review, we summarize published findings of epigenetic responses to replicative, oxidative, mechanical, and inflammatory stress on various types of adult stem cells.**

## Introduction

### What are stem cells?

Stem cells are defined as cells with unique abilities for repairing and maintaining functional tissues throughout the body via differentiation and self-renewal. They are found to exist in both embryonic and adult tissues, in which different types of stem cells are defined based on their differentiation potential and the tissue they reside in.

Embryonic stem cells primarily exist during early development and are critical for giving rise to more specialized cells in the body. These stem cells can be classified as either totipotent or pluripotent [1, 2]. Totipotent stem cells can differentiate into extraembryonic tissues such as the placenta and all specialized cells, one example being zygotes. They exist only during early embryonic development. Pluripotent embryonic stem cells are capable only of differentiating into the three primary germ layers and are found during embryonic development in the inner mass of blastocysts.

Induced pluripotent stem cells generated by reprogramming adult cells are like these embryonic pluripotent cells [3, 4]. Adult stem cells exist in small populations within specific niches of functional tissues, such as BM, adipose tissue, or skeletal muscle. Each adult stem cell population displays different behaviors, morphology, and gene expression profiles to meet the demands of the tissue they serve. When compared with embryonic stem cells, their differentiating potential is limited to one or more cell lineages; however, these stem cells are also capable of self-renewal [5].

### What is stress?

Stress as a concept has been used in various fields and has evolved and been expanded significantly in recent times (reviewed by reference 6). For the purposes of this review, we will define stress as a state when biological, physiological, or psychological homeostasis has been challenged by a condition (stressor) and requires compensatory responses (stress response) to prevent an accumulation of acquired insults to reestablish normal conditions. However, when a stressor is introduced at a low enough intensity, the body may gain some form of benefit such as a more robust tolerance to varying stressors, an overall reduction in all-cause mortality, or anabolic muscle growth for increases in raw strength; of course, dependent on the stressor introduced [7, 8, 9, 10, 11]. This form of stress is known as eustress, and the biphasic dose response has been defined as hormesis [12]. The hormetic effects of varying stressors, however, typically take place in controlled environments (e.g., saunas and curated diets) or survival-based situations (e.g., food scarcity). Thus, stress is often experienced negatively when resulting in an acute or chronic accumulation of insults to the organism. Stress can be experienced on varying levels—such as systemic and (sub)cellular. Cellular stress affects the cell body, proteomic, genomic, and epigenomic levels, and is elicited by stressors originating extra- or intracellularly. In this review, we will mainly focus on the stressors involved in cellular stress.

Extrinsic cellular stresses are often caused by external events the organism is exposed to and may often result in hormetic effects under the appropriate intensities. One example that has gained recent attention within the scientific and health/fitness communities

---

Epigenetics and Stem Cell Unit, Translational Gerontology Branch, National Institute on Aging, Baltimore, MD, USA

Correspondence: isabel.beerman@nih.gov
*Joey Llewellyn and Rithvik Baratam contributed equally to this work

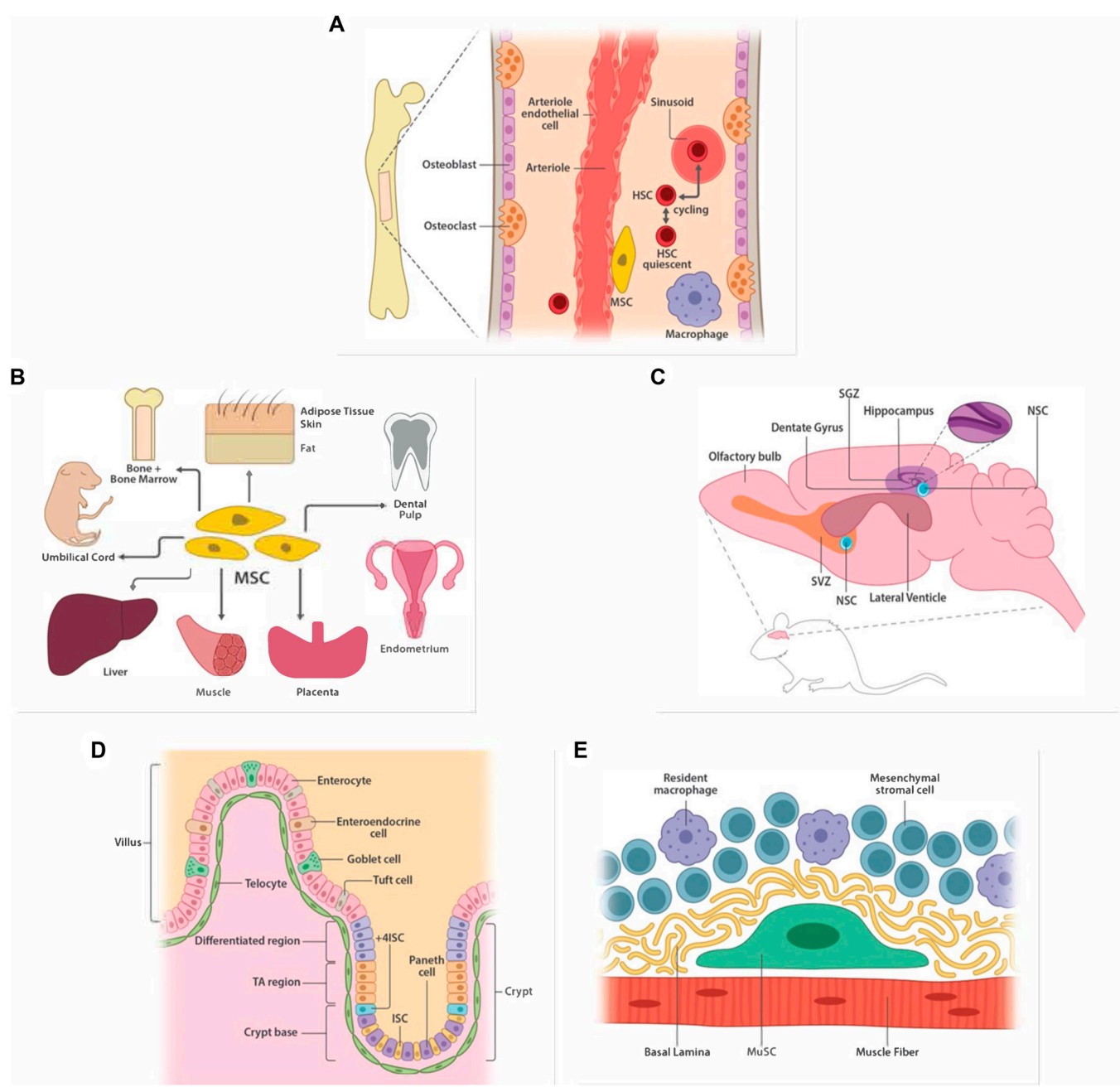

**Figure 1. Stem cells and their niche.**
**(A)** Hematopoietic stem cells (HSCs) are largely found residing in the bone marrow microenvironment. Cycling HSCs are found to localize near the sinusoid whereas those that are quiescent tend to localize closer to the endosteal (bone-lining) wall. Cells neighboring HSCs in the bone marrow (e.g., mesenchymal stem cells [MSCs] and arteriole endothelial cells) often hold roles in regulating HSC function and maintenance via the secretion of cytokines, growth factors, and other small molecules. **(B)** Mesenchymal/stromal stem cells (MSCs) are found to reside in several adult (adipose tissue, compact bone, bone marrow, dental pulp, liver) and embryonic (endometrium, placenta, umbilical cord) tissues. MSCs directly regulate cellular function and maintenance of nearby cells within their respective niches, including adult stem cells, through several secretory mechanisms. **(C)** The existence of neural stem cells (NSCs) has been established in rodents in the subgranular zone of the hippocampal dentate gyrus and the subventricular zone of the lateral ventricles. Subventricular zone NSCs produce (inhibitory) oligodendrocytes and interneurons, whereas subgranular zone NSCs generate (excitatory) granule neurons. The controversy regarding the existence and activity of NSCs in humans has not yet been fully resolved. **(D)** The niche residency of intestinal stem cells (ISCs) has been established largely within the crypt base (below the villi) along the intestinal epithelium. Their location has been long debated; however, their exact position along the crypt base has yet to be fully resolved. ISCs hold multipotent differentiation capacity, where ISCs adjacent to longer villi are biased towards producing enterocytes whereas those adjacent to shorter villi will largely produce secretory cells (e.g., goblet cells). **(E)** Muscle/satellite stem cells (MuSCs) are uniquely identified by their anatomical location between the basal lamina and myofiber of skeletal muscle tissue. Unlike other adult stem cell niches, the MuSC local microenvironment is anatomically dynamic during muscle regeneration and attributable to regulatory networks by MuSCs and other functionally diverse resident cells.

over the past decade is heat stress and its robust induction of the heat shock response (13). When an organism is exposed to mild heat stress an evolutionarily conserved family of proteins called heat shock proteins (HSPs) are robustly expressed, relative to intracellular baseline levels, and localize within the cytoplasm and various organelles of the cell. Once activated, HSPs act as chaperones to participate in native protein quality management and stabilization if proteins are misfolded, denatured, or have aggregated during exposure to said stress. Interestingly this pathway is also activated by other forms of stress (14), and if unmanaged, proteins that have entered a de-stabilized conformation will have their normal function impaired and become more vulnerable to additional changes that could lead to cell damage and ultimately death (15, 16, 17).

Intrinsic cellular stress is unfavorable condition caused by several cellular processes either in conjunction or independent of one another. For example, oxidative stress arises from within the cellular microenvironment and is caused by an imbalance between the production of reactive oxygen species (ROS) in cells and the capacity of the cellular system to clear out these oxidative free radicals (18). This imbalance is generally known to be detrimental to cells in which an accumulation of insults such as DNA damage and inflammation may lead to malignancies such as cancer and cardiovascular diseases (19). These forms of stress, and others, will be discussed below.

## Stem cells

### Hematopoietic stem cell (HSC)

HSCs are defined by their lifelong self-renewal capacity and multipotency to differentiate into all cell types that comprise the blood and immune systems. HSCs are a heterogeneous population of cells mostly found in the BM (Fig 1A) with potential to generate a large repertoire of cells, ranging from erythrocytes (red blood cells) that carry oxygen, thrombocytes (platelets) which facilitate coagulation factors for clotting, and cells of the innate and adaptive/learned immune systems such as granulocytes, monocytes, natural killer (NK) cells, B cells, and T cells which recognize and fight off foreign substances (20, 21, 22, 23). Many of these cells also have a high turnover rate, with 86% of human cell turnover coming from red blood cells and neutrophils suggesting robust maintenance of the HSC compartment must occur throughout life to ensure continued production of these cells (24). Numerous studies in mouse and human models have shown the multipotent differentiation and self-renewal capacity of HSCs is critical for maintaining the blood and immune systems throughout an organism's lifespan with a loss of this tight regulation leading to maladies such as BM failure, stem cell exhaustion, and leukemia (25, 26, 27, 28, 29). HSCs are maintained in a quiescent state, where they remain metabolically conservative, possibly reducing the induction of intracellular stresses and potential insults to their function (30). Tight regulation of cell-cycle entry is critical for HSC preservation, where both cell-intrinsic and -extrinsic regulators are involved. Extrinsic regulation includes factors secreted from the BM niche, of which neighboring cells (mesenchymal stem cells [MSC], adipocytes, osteoblasts, perivascular cells, endothelial cells, arterioles, and sympathetic nervous neurons) secrete adhesion molecules, cytokines, growth factors, and other proteins into the ECM that either drive HSC cell-cycle entry or revert it back to quiescence, aiding in stem cell function or maintenance, respectively (Fig 1A) (31, 32, 33, 34, 35).

To gain a multidisciplinary understanding of the complex system and balance of human HSC biology, mouse models have been widely used ever since the early 1960s when HSCs were first conceptualized from BM transplantation assays (36). Advancements made in the technology and methodologies for transplantation studies, ex-vivo/in-vivo culturing, gene expression manipulation, and other assays performed on mice allowed us to gain more targeted and high-throughput insight into human HSC function and maintenance; this insight demonstrated conservation of HSC function and maintenance between mice and humans. However, it has also been revealed that there are numerous critical differences between human and mouse HSCs regarding phenotype, function, and regulation of maintenance and multilineage differentiation that cannot be overlooked. One area of importance involves the immunophenotypic characterization of HSCs and hematopoietic progenitor cell, surface markers, which they differ between humans and mice. These differences in immunotypic profiles also confer functional differences despite both human and murine HSCs having similar mechanisms and phenotypes. For example, Flt-3 is also expressed in mice, but its importance lies in regulating lymphopoiesis, not HSC maintenance or function when compared with human HSCs (37). In murine HSCs, c-Kit provides a higher degree of survival than Flt-3 whereas the opposite holds true for human HSCs (38). One of the most striking differences between human and mouse immunophenotypic markers is CD34. In the murine system, HSCs are classified as CD34$^-$, whereas human HSCs are characterized as CD34$^+$. Intriguingly, a population of CD34$^-$ human HSCs has recently been identified in the cord blood (39); however, it is unclear if this population of megakaryocyte/erythrocyte-biased HSCs also exists in adult BM.

### Mesenchymal stem/stromal cell (MSC)

MSCs are a population of cells residing mostly in the BM and connective tissues (Fig 1B) that manage the production of numerous stromal tissue-type lineages (40). These include adipocytes (fat cells), hypertrophic chondrocytes (cartilage cells), stromal cells, tendon/ligament fibroblasts, myocytes (muscle cells), and osteoblasts (bone cells). One misconception relating to its stem cell identity is that MSCs are believed to be equivalent regardless of their tissue of origin. As reviewed by Phinney and Sensebé (41), not all MSCs originating from several distinct adult tissues are equivalent despite sharing similar genome expression profiles and phenotypes. In recent years, in vitro and in vivo studies comparing MSCs from different tissues further demonstrated the differences in several cellular facets, such as immunophenotypic markers, growth kinetics, and secretome profiles (42, 43).

One important property of MSCs is their ability to immediately respond to signals because of increased stromal tissue demands and mount the appropriate regenerative mechanisms (44). This is in part because of its homing ability allowing these cells to migrate to sites of injury or inflammation. Once MSCs have reached their target site, they actively differentiate to produce the appropriate cell type/s (45). An additional property of MSCs regarding their therapeutic potential is their immunomodulatory properties by which

these cells are able to suppress immune responses through the secretion of extracellular vesicles containing anti-inflammatory cytokines and direct interaction with immune cells (46).

MSCs can be characterized by their ability to adhere to plastic surfaces, form colonies, and differentiate into adipocytes, chondroblasts, or osteoblasts in vitro (47). In addition, MSCs can be identified by a combination of cell surface markers. Human MSCs (hMSCs) express CD73, CD90, and CD105, although it lack the expression of a pan hematopoietic marker CD45, CD11b, CD14, CD19, CD45, CD79a, and HLA-DR (48). Mouse MSCs are identified by their expression of Sca1, CD29, CD44, CD73, CD105, CD106. It is important to note that a unique panel of markers unequivocally defining MSCs is still being developed, so in addition to a combination of cell surface markers, functional assays (e.g., differentiation and immunomodulatory assays) are often used to establish the stemness of the population (49, 50).

The therapeutic potential of MSCs in various disease models and injuries has been studied in both human and murine models. Their regenerative capacity in bone and tissue repair exhibits promising therapeutic potential because of their availability, multilineage differentiation capacity and immunomodulatory effects, as shown in autologous and allogenic MSC-based transplantation studies (51, 52, 53, 54, 55, 56, 57). However, their clinical applications are still an area of research and debate, considering that several challenges remain such as the optimization of isolation and expansion protocols and long-term effects, among others (58).

### Neural stem cell (NSC)

Whereas the existence and activity of NSCs have been established in many animals, from mice (59) to macaques (60, 61), there remains a bit of controversy about whether neurogenesis, the process of generating new neurons from NSCs, occurs in adult humans. In short, certain studies using an array of methods (ranging from radioactive carbon-based cell-birth dating to immunohistochemistry) found evidence of ongoing neurogenesis up to the ninth decade of life, with a modest age-associated decline (62, 63, 64, 65). Conversely, some studies that used immunohistochemistry or single-nucleus RNA-seq found no robust evidence of adult neurogenesis in humans (66, 67). Until this debate is resolved through advances in technology (e.g., magnetic resonance spectroscopic imaging, more cells profiled by sc/sn RNA-seq, etc.) and/or in tissue processing (fixation, short postmortem delay, epitope retrieval, etc.), the studies mentioned above are the strongest proof of persistent neurogenesis in adult humans.

Neurogenesis itself is considered to occur in two canonical areas of the brain—the subgranular zone (SGZ) of the hippocampal dentate gyrus (DG) and the subventricular zone of the lateral ventricles (Fig 1C), with an age-associated decline (68). We will just briefly mention that adult neurogenesis has been reported in some other brain areas as well—from the neocortex (69) and the cerebellum (70) to the amygdala (71), but will not focus on them in this review. Adult neurogenesis can be broadly divided into several distinct stages: (1) proliferation—division of progenitor cells, (2) differentiation—selection of the neuronal fate of newborn daughter cells, and (3) survival—long-lasting incorporation of cells into the circuitry. Given that NSCs are the source of new neurons in the adult brain, they represent a potential therapeutic target for numerous disorders, ranging from mood disorders such as major depression, to neurodegenerative diseases such as Alzheimer's disease, which are usually associated with deficits in neurogenesis (62, 72, 73, 74).

Adult NSCs are multipotent, they generate lineage-specific cell types which are restricted based on the neurogenic area they belong to (75), similar to MSCs. Specifically, in the SGZ, NSCs generate cells that are committed to excitatory granule neuron and astrocytic fates (76), whereas in the subventricular zone, they produce inhibitory oligodendrocytes and interneurons (77) (Table 1). The quiescent NSCs in the SGZ are also called type-1 radial glia-like cells (RGLs) and can be further split into type α cells and type β cells based on their morphology and response to pro-proliferative stimuli. It is the type α cells that give rise to granule neurons and astrocytes, whereas type β cells do not proliferate and their identity is less clear. Because type β cells express both stem cell markers (e.g., Nestin, Sox2) and mature astrocytic markers such as S100β, it is hypothesized that they may be in an intermediate state between quiescent RGLs and astrocytes (78). When the quiescent RGLs are activated, there are two paths they can follow—(a) symmetrical or (b) asymmetrical division. Path (a) leads to self-renewal via generation of new RGLs, whereas (b) leads to proliferating intermediate progenitor cells, or type-2 cells, which can be further subdivided to type-2a and type-2b cells. Type-2a cells continue to express Sox2, an NSC marker, whereas type-2b cells express DCX, usually used as a marker of immature neurons (79). These type-2 cells give rise to bipolar neuroblasts (type-3 cells) which become immature neurons, and subsequently, a subset of them (50% in rats) (80) will become mature and integrated granule neurons with axonal projections into the CA3 region of the hippocampus. These newborn neurons have multiple important roles, ranging from pattern separation (81, 82), resilience to and remission from stress (83, 84, 85) to learning and memory (86, 87). Whereas the effects of regulators of NSC proliferation in the context of aging have recently been summarized elsewhere (88), in this review, we will describe the effects of stressors on NSCs.

### Intestinal stem cell (ISC)

ISCs differentiate into all specialized cells making up the intestinal epithelium belonging to the absorptive (enterocytes) or secretory (goblet, entero-endocrine, tuft, and Paneth cells) lineages. The intestinal epithelium is lined with a continuum of crypts and villi that is required to rapidly turnover every few days (127). To maintain normal gut function in the absorption of important macro- and micronutrients from digested foods and symbiotic relationships with microbes, ISCs are biased towards the absorptive lineage, where enterocytes comprise roughly 90% of total terminally differentiated cells; cells downstream of the secretory lineage comprise of roughly 1–10% (128). The organization of the intestinal tract is such that varying sections house villi of different lengths, to which longer villi will typically contain enterocytes for digestion and absorption functions and shorter villi containing mostly goblet cells for the secretion of mucus to provide lubrication for the passage of undigested food towards the colon. An example of this is the duodenum which houses longer villi, of which digestion and absorption of food takes place and is facilitated by enterocytes.

**Table 1. Summary of epigenetic responses to stress in adult stem cells.**

| Adult stem cell type | Epigenetic response to stresses | | | |
| --- | --- | --- | --- | --- |
| | replicative stress | oxidative stress | mechanical stress | inflammatory stress |
| Hematopoietic stem cell (HSC) | ↓ histone methylation (89), histone H4 deacetylation (90) | ↑ [a]histone H2A monoubiquitination (91), histone H3 mono/di/tri methylation (91), DNA hypomethylation of regions involved in differentiation potential (92) | N/A | ↓ histone demethylases: UTX/KDM6A (93), DNA methylases (94), DNA demethylases (95) |
| | ↑ histone modifications: γH2AX (96), H3K4me3 (90), acetylation (97), [a]chromatin accessibility (89) | | | ↑ chromatin remodeling (98, 99), DNA methylation: IgK locus (98, 99) |
| Mesenchymal/stromal stem cell (MSC) | ↓ heterochromatin (100) | ↑ [a]γH2AX (101, 102), miRNA-210 (103), miRNA-29a-3p (104), miRNA-30c-5p (104), DNA hypermethylation (104), mtDNA hypermethylation (105), histone H3K27me3 (106) | ↓ DNA methylation of adipo- and osteogenic promoting regions (107) | ↑ DNA hypermethylation (108), histone methylation at H3K9 (109), DNA hypomethylation (110), histone variant mH2A1.1 (111) |
| | | | ↑ histone H3 deacetylation of osteogenic promoter region of Jag1 (112) | |
| Neural stem cell (NSC) | N/A | N/A | N/A | ↑ histone demethylation at H3K9 (113) |
| Intestinal stem cell (ISC) | ↑ DNA methylation of promoter regions marked with H3K27me3 (114), genome-wide DNA methylation (115) | ↑ [a]histone modification: γH2AvD—analog to γH2AX (116) | N/A | ↑ heterochromatin (117), histone modifications: H2A.Z (118) |
| Muscle/satellite stem cells (MuSC) | N/A | ↑ heterochromatin of ink4a locus (119) | ↓ H4R3 dimethylation of Dnmt3b and p21 promoter regions, [a]H3K4 trimethylation of Dnmt3b promoter region (120, 121, 122, 123, 124) | ↓ histone methylation at H4K20 (125) |
| | | | ↑ [a]H3R8 dimethylation of upstream enhancer-like region and p53 binding sites, [a]DNA hypermethylation of p21 promoter region, [a]H3K4 trimethylation of p21 promoter region (120, 121, 122, 123, 124) | ↑ histone methylation at H3K27 (126) |

[a]requires further investigation to confirm.

With each subsequent differentiation, the progeny of ISCs migrate along the crypt-villus axis. As such, the crypt-villus axis holds three distinguished zones based on the degree of differentiation of ISCs moving upwards from the crypt base and along the villus: the stem cell zone, transit-amplifying (TA) zone, and mature cell zone (Fig 1D). ISCs within basal crypt niches undergo symmetric cell division, where each division results in two identical daughter cells, either both retaining stemness or both committing to differentiation. This process ensures a balanced production of new stem cells and differentiated progeny, maintaining tissue homeostasis. Within the niche, ISCs experience neutral competition for space, where central ISCs, located deeper in the crypt, have a positional advantage, increasing their likelihood of self-renewal compared with border ISCs, which are more prone to displacement and differentiation (129). This spatial hierarchy and competitive dynamic result in a stochastic pattern of ISC turnover, where over time, individual crypts drift towards monoclonality, ultimately being maintained by a single ISC lineage (130). Through symmetric cell division, ISCs give rise to rapidly cycling daughter cells known as TA cells will then undergo a decrease in replication to terminally differentiate into cells of the intestinal epithelium (131, 132, 133).

The characterization of ISC identity using cell surface markers has been established for both humans and mice (134). Human ISCs may be identified as CD44+CD24loCD166+GRP78lo/−c-Kit−. Similarly, mouse ISCs may be identified as CD44+CD24−/loCD166+. ISC niche residency is a topic that has been long debated for the past few decades. There are currently two competing models: the "stem cell zone" model (135) and the "+4" model (136).

The "stem cell zone" model first proposed the identification of cell-cycle active crypt base columnar (CBC) cells containing phagocytic activity with the purpose of clearing dead cells at the base of intestinal crypts as shown in irradiated mice containing radiolabeled phagosomes (137). CBC cells were also thought to be ancestors to specialized cells of the four major ISC lineages. It was not until the late 1990s that this model was further refined through clonal analysis of intestinal epithelial progenitor cells involving the induction of gene mutation for heritable cellular phenotypes within crypts (135). This demonstrated that cells of the four major lineages contained the unique mutation, strongly suggesting the presence of a population of multipotent stem cells. This model became more established upon the identification of the CBC cell-specific marker *lgr5* (leucine-rich repeat-containing G protein-coupled receptor 5),

a gene target of the WNT signaling, in mouse models of both in vivo and ex vivo assays (138, 139).

The "+4" model was first postulated in the late 1970s in mouse models of cell-tracking assays incorporating radiolabeled nucleotides in intestinal epithelial cells exposed to different forms of irradiation (136). Epithelial cells at varying positions along the intestinal crypt were examined; it was discovered that endothelial cells around the +4 position of the crypt (above the Paneth cell compartment) (Fig 1D) were found to retain radiolabeled nucleotides and were the most radiosensitive, showing typical behavior of quiescent/slow replicating cells and stem cells, respectively. This model was further established in the 2000s with the discovery of the +4 stem cell marker *bmi1* (a component of the polycomb repressive complex 1 [PRC1]) (140, 141), *Hopx* (homeodomain-containing protein) (142), *Lrig1* (leucine-rich repeats and immunoglobulin-like domains 1) (143), *Tert* (telomerase reverse transcriptase) (144), and others. However, most of these stem cell markers were found to be unreliable in defining an additional stem cell compartment aside from LGR5+ CBC cells shown by single-molecule mRNA FISH in mouse intestinal sections; this assay displayed a broad expression for these markers within other specialized cells, such as *Dcamkl1* tuft cells (145). Nonetheless, these models have characterized a population of stem cell-like cells residing around the +4 position of the intestinal crypt that contributes to intestinal regeneration.

### Muscle stem cell (MuSC)

MuSCs, also known as satellite cells, are a type of adult stem cell that resides in skeletal muscle tissue (146) (Fig 1E). These cells play an active role in muscle growth, maintenance, and repair under homeostatic conditions and in response to injury.

MuSCs are small, quiescent cells located between the basal lamina and the sarcolemma of skeletal muscle fibers (147). Human MuSCs are characterized by their expression of several markers including Pax7, M-cadherin, α7-integrin, CD56, CD82, and CD318 (148). Mouse MuSCs may be characterized and isolated by their expression of β1-integrin, CXCR4, VCam1, α7-integrin, and CD34 with Sca1, CD31, CD45, Mac1, and Ter119 exclusion (149). Mouse MuSCs are also identified by Pax1 expression; however, it is not used for its isolation. One of the unique characteristics of MuSCs is their ability to self-renew and differentiate into myocytes (muscle cells). Upon activation, MuSCs exit quiescence and undergo proliferation, generating a pool of myoblasts that can differentiate into mature muscle fibers. MuSCs also can regenerate themselves through asymmetric division, in which one daughter cell remains a stem cell, whereas the other differentiates into a myoblast. This ensures that the MuSC pool is maintained throughout an organism's lifetime.

MuSCs play a critical role in muscle growth, maintenance, and repair. During muscle growth, MuSCs proliferate and differentiate into myoblasts, which fuse with existing muscle fibers to increase muscle mass. Whereas MuSCs are important for muscle growth, MuSCs may not be needed for muscle maintenance. Genetic deletion of Myf6 in skeletal muscles leads to the exhaustion of the MuSC pool in adult mice. However, despite the depletion of MuSCs, muscle differentiation remains unaffected, indicating that Myf6 specifically influences the stem cell population rather than the differentiation process itself (150). MuSCs remain quiescent but are poised to activate and repair muscle tissue in response to injury or damage. When activated, MuSCs proliferate and generate mature muscle fibers. This process is tightly regulated by various signaling pathways, including Notch, Wnt, and TGF-β. Dysregulation of these pathways can lead to impaired muscle regeneration and muscle-wasting conditions such as muscular dystrophy.

The activity of MuSCs is tightly regulated by various signaling pathways and transcription factors (147). One of the key regulators of MuSCs is the transcription factor Pax7. Loss of Pax7 results in a depletion of MuSCs and impaired muscle regeneration. In addition to Pax7, Notch signaling is required for the activation of MuSCs and the generation of myoblasts. Wnt signaling also promotes the proliferation and differentiation of MuSCs, whereas TGF-β signaling inhibits MuSC activation and promotes fibrosis (the development of mature myofibers).

# Stress-Induced Epigenetic Alterations

The broad mechanisms involving the transfer of genetic information to the production and modification of the biochemically functional proteins involve transcription, translation, and protein modification. To allow for specialized cells to arise, intra-/intercellular networks to form, and various cellular functions to run efficiently, mechanisms for tight regulation of gene expression must be established. Epigenetics is defined by heritable and stable chemical alterations imposed on the chromatin landscape and gene silencing without altering the DNA sequence. The epigenetic mechanisms involved establish a critical layer of control, predominantly within the nucleus, that overall regulates gene expression and silencing. These mechanisms are classified into four major classes: histone modification, chromatin accessibility, DNA methylation, and noncoding RNA.

Cellular stress can be generally defined as an event causing and/or condition resulting in an acute or chronic shift from a physiological homeostatic state in which a given cell optimally thrives. Intrinsic cellular responses will then act accordingly to restore homeostasis, preventing potential macromolecular damage performed onto DNA, RNA, proteins, and lipids, as well as other insults, that could result in short-/long-term effects detrimental to the function of a cell. It is important to note that depending on the type of stress, degree of insult, and adaptive capacity of the cell, specific defensive mechanisms may be mounted and could induce hormetic effects. One example involves the intracellular accumulation of mis-/unfolded proteins because of various stresses such as oxidative stress and heat shock that could lead to cell death. To address this, increased activation of chaperone proteins (e.g., hsp70) and the unfolded protein response pathway are mounted as a cytoprotective mechanism (151). For the focus of this review, we will cover a handful of the major stresses involved in several hallmarks of aging within stem cells: replicative stress, oxidative stress, mechanical stress, and inflammatory stress.

### Replicative stress

Replication stress is caused by the formation of interstrand cross-links during DNA replication and is characterized by replication fork

stalling or a slowdown in DNA synthesis, ultimately resulting in genome instability, the progressive accrual of DNA damage, and the development of various diseases. These interstrand cross-links, which covalently link the two strands of the DNA helix, prevent the normal unwinding of DNA required for replication and transcription. Replication stress is caused by DNA lesions or obstacles induced by endogenous agents such as ROS and metabolic byproducts, or exogenous agents such as UV radiation, chemical exposure, and certain chemotherapy drugs. The presence of these obstacles disrupts the normal progression of the replication machinery, leading to the formation of single-stranded DNA regions and the activation of DNA damage response (DDR) pathways. Over time, the inability to efficiently resolve replication stress can contribute to chromosomal aberrations, mutations, and an increased risk of oncogenesis and other genetic disorders.

The transition from quiescence to the cell cycle in stem cells is critically affected by epigenetic changes that become dysregulated during aging. For example, a study identifies a marked increase in H3K4me3 at the promoter and first exon of the Hoxa9 gene in aged HSCs, driven by the recruitment of the Mll1 complex and its scaffold protein Wdr5 (152). This epigenetic alteration leads to the aberrant activation of developmental pathways, including Wnt, TGF-β, and JAK/STAT signaling, which are inhibitors of satellite cell function in aging muscle. These changes are linked to increased DNA replication stress, as the overexpression of Hoxa9 in young cells mimics aging-associated defects, such as heightened apoptosis and reduced cell proliferation. This overexpression also suppresses cell-cycle regulators and induces senescence genes.

### HSC

The lifelong self-renewal capacity of HSCs is crucial for blood regeneration and giving rise to immune cells. This is supported by their quiescent state, where HSCs remain dormant and are metabolically conservative, thus reducing the induction of intracellular stress. When needed, HSCs are required to respond quickly and efficiently to a given situation where increased blood demands and/or cellular output for various immune responses, such as inflammation caused by bacterial infection. With the ability to rapidly jumpstart the appropriate intracellular processes for myeloid- or lymphoid-directed differentiation, comes a price that exposes HSCs to replication stress considering the immediate demand for DNA replication and high turnover rate. This is why it is proposed that the stem cells will differentiate to a progenitor cell without self-renewing potential and the bulk of the demand is met by increasing progenitor cells when protecting the stem cell population. However, when HSCs experience acute/chronic replicative stress, conserved response pathways such as the ATR-mediated DDR and ATM-mediated apoptotic priming pathways are activated to prevent and resolve genotoxic insults that arise (96, 153, 154, 155). Epigenetic alterations and responses, such as histone modifications, have been studied to better understand and elucidate the cellular defense mechanisms that HSCs mount to resolve replication stress and the potential for insults negatively impacting genome stability and cell survival.

In HSCs from young (6–12 wk) and old (22–30 mo) C57BL/6 mice, a study compared the prevalence and impact of replicative stress in an age-dependent manner and investigated a novel function of

yH2AX as an epigenetic regulator (29). Old cycling HSCs under ionizing-radiation-induced replicative stress in vitro displayed an accumulation of yH2AX foci without DDR activation accompanied by impaired S phase progression of the cell cycle, increase in senescence-associated β-galactosidase (SA-β-Gal) and Cdkn2a (p16) expression, decrease in DNA helicase components MCM4 and MCM6 gene/protein expression, and presence of chromosomal breaks. Induced replicative stress in young HSCs displayed similar effects and functional decline as old HSCs in vitro or after BM transplantation, however, to a lesser extent. To further elucidate the molecular mechanism(s) driving the observed effects, this study also demonstrated that yH2AX signals accumulated on ribosomal DNA (rDNA) in the nucleolus and was associated with decreased 47S rRNA transcripts and ribosomal biogenesis in quiescent old HSCs. This was not a result of replicative stress-induced mutagenic impact on rDNA, rather, it is suggested that nucleolar yH2AX acts as an epigenetic histone modification that represses rDNA gene expression because of failed dephosphorylation by yH2AX phosphatase PP4c as it was mis-localized in quiescent old HSCs, primarily being found in the cytoplasm. PP4c was found in both the nucleus and cytoplasm of cycling and quiescent young HSCs under replicative stress. Persistent yH2AX that arise from replicative stress thus may be a long-term or permanent histone modification silencing rDNA and downstream translation of critical regulators of HSC function and maintenance.

*Ott1* (also known as RNA Binding Motif 15 [RBM15]), holds broad regulatory roles in hematopoiesis and is required to preserve HSC quiescence during replicative stress, but not steady-state conditions. This was found in *Ott1* KO mice where HSCs derived from young mice displayed premature aging phenotypes such as higher sensitivity to replicative stress, increased myeloid lineage output bias, and increased expansion (90). *Ott1* KO HSCs were also found to have a significant reduction in quiescent cells. Part of the protective role of *Ott1* in replicative stress is its regulation of the Thpo/c-Mpl axis via c-Mpl H4 deacetylation and H3K4me3 involving histone deacetylase and methyltransferase Hdac3 and Setd1b, respectively (156) (Fig 2A). c-Mpl is a transmembrane hematopoietic cytokine receptor promoting quiescence or proliferation of HSCs in a Thpo concentration-dependent manner, to which it is *Ott1*-dependent.

Polycomb group (PcG) proteins are well known as epigenetic histone modifiers (157) and are found to be involved in preventing and during replicative stress response (158, 159, 160). Enhancer of zeste homolog 2 (EZH2) is a PcG protein and well-characterized histone methyltransferase known for its role as a transcriptional repressor (89, 161). It is thought that replicative stress contributes to chromatin instability via the loss of histone methylation motifs that facilitate DNA condensation, among other factors (162). EZH2 was found to be heavily down-regulated in murine HSCs experiencing severe replicative stress because of serial transplantations when displaying signs of stem cell exhaustion (163). When EZH2 was overexpressed in exhausted HSCs, long-term self-renewal and fitness was restored, as was repopulation capacity.

The CREB-binding protein (CBP), and its paralog p300, have been well defined as crucial transcriptional coactivators and histone acetyltransferases in a wide array of cellular processes in development and adult tissue homeostasis such as stem cell differentiation, proliferation, and apoptosis (164, 165, 166, 167, 168). CBP

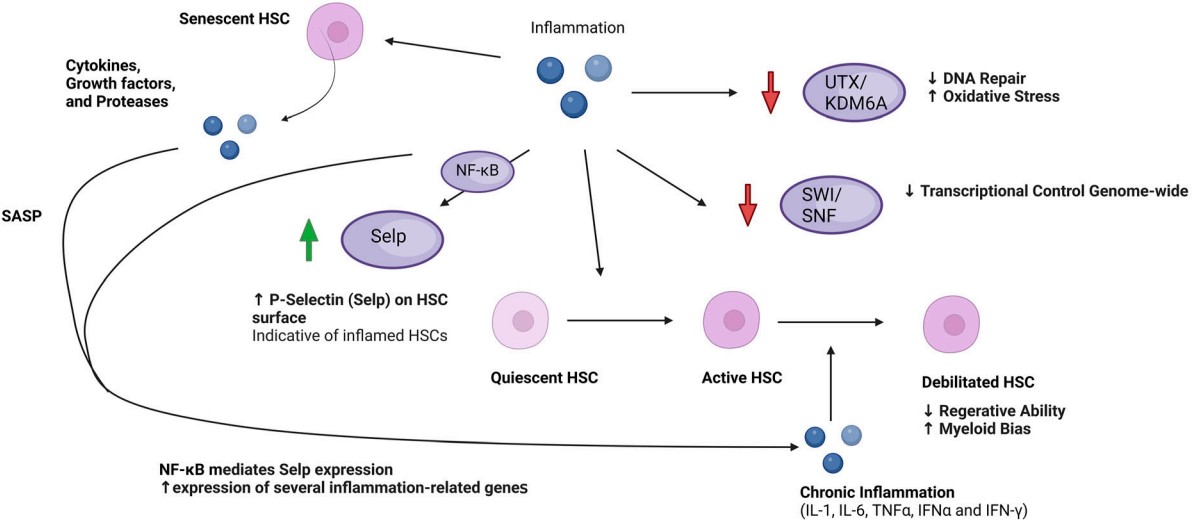

**Figure 2. A schematic illustration of epigenetic changes in HSCs because of various stressors.**
**(A)** HSCs under IR-induced replicative stress displayed an accumulation of yH2AX foci without DDR activation accompanied by impaired S phase progression, senescence markers, and a decrease in DNA helicase components. Ott1 maintains HSC quiescence during replicative stress, where HSCs exhibit premature aging phenotypes. EZH2 is down-regulated in murine HSCs under severe replicative stress, leading to stem cell exhaustion. CBP is a transcriptional coactivator and histone acetyltransferase involved in several processes, such as differentiation, proliferation, and gene regulation. **(B)** PRC1 and PRC2 complexes are key epigenetic regulators in gene silencing via histone modifications. Autophagy regulates oxidative stress by clearing mitochondria, but its repression increases hypomethylation in genes that accelerate myeloid differentiation. **(C)** Inflammatory signals reduce expression of the histone demethylase UTX/KDM6A, impairing DNA repair and increasing oxidative stress. Global gene expression analyses have identified down-regulation of SWI/SNF-related genes, which are involved in chromatin remodeling and transcriptional silencing. Inflamed HSCs show increased expression of P-selectin (Selp), driven by heightened NF-κB pathway activity, which up-regulates inflammation-related genes and contributes to HSC functional decline. Chronic inflammation also induces a senescence-associated secretory phenotype in HSCs, thereby perpetuating inflammation and impairing HSC function.

regulates gene expression through two different mechanisms: (1) by acting as a scaffold for the recruitment of transcriptional complexes to RNA polymerase machinery and (2) direct involvement in

protein and histone lysine acetylation (97, 164, 169). In HSCs derived from young *Cbp* conditional KO mice, conditional ablation of CBP was found to result in overall dysregulation of HSC homeostasis

and reconstitution capacity when met with replicative stress as observed in a BM transplantation assay (170). These HSCs were found to display aberrant regulation of the cell cycle, where stem and progenitor cells largely remained in the quiescent state, and increased apoptosis in the LSK population, contributing to cellular exhaustion. Furthermore, a genome-wide analysis showed that CBP was bound to loci regions involved in several critical HSC transcriptional networks/regulators, including the Heptad network, members of the Kruppel-like factor family (*Klf2*, *Klf10*, and *Klf13*), *Runx1*, *Gfi1b*, and *Gata2* (171). This was strongly associated with the down-regulation of Heptad target genes involved in HSC differentiation, cell-cycle regulation, and self-renewal in *Cbp* KO mice and overlap in genes bound by CBP and Heptad factors. This demonstrates the direct regulation of CBP on critical HSC-specific transcriptional networks that are critical for ensuring HSC survival and maintenance under stressed conditions. Considering its epigenetic function, CBP may modulate these networks and corresponding transcription factors via histone and protein acetylation. However, additional studies investigating the epigenetic influence and long-term alterations CBP has on HSCs in these networks and factors would need to be performed.

The current studies show that replicative stress can activate several factors to elicit epigenetic alterations primarily as histone modifiers for preserving HSC quiescence and normal function (Table 1). The roles of several histone modifiers previously mentioned hold cytoprotective roles in HSCs against phenotypes like what is seen in an aging context, namely increased expansion and myeloid-biased differentiation. Replicative stress-induced down-regulation of histone methyltransferase EZH2 led to chromatin instability via resulted HSC exhaustion. These histone modifiers are also involved in evolutionarily conserved replicative stress response pathways, such as DDR pathways. To our knowledge, there are no published studies reporting the effects of replicative stress on other facets of the epigenetic landscape, namely DNA methylation and noncoding RNAs, but these would be an area of interest to further investigate.

### MSC

MSCs have not been reported to have extensive replicative stress, although autologous MSC-based studies have elucidated a protective role that epigenetic alterations have in aging MSCs in response to stress and other negative aging-associated phenotypes.

One of the common phenotypes of replication stress in adult somatic cells is cellular exhaustion induced by age-associated ER stress caused by impaired proteostasis via the accumulation of un- and misfolded proteins. This has been primarily observed in hMSCs in natural and premature aging disease models (100, 172). As a consequence of age-associated cell exhaustion and ER stress, organelle homeostasis becomes disrupted considering the ER is connected to or in contact with membrane-bound organelles such as the nuclear envelope (NE) or Golgi apparatus and mitochondria. However, the exact causal molecular mechanisms have yet to be fully understood. In vitro functional assays involving hMSCs demonstrated that ER stress sensing/ER transmembrane protein activating transcription factor 6 (ATF6) maintains organelle homeostasis and protects hMSCs directly from ER and, arguably, replicative stress (173). This was shown in CRISPR/Cas9-mediated

$ATF6^{-/-}$–induced hMSCs from hESCs, where hMSCs displayed accelerated cellular senescence, structural dysregulation in the ER, NE, mitochondria, and heterochromatin. These effects were also present in FOS-deficient hMSCs, suggesting a critical role for FOS as a downstream mediator of ATF6 in ER stress response. *FOS* is a novel ATF6 target gene in hMSCs that encodes for a leucine zipper protein forming a transcriptional complex (AP-1) with factors of the JUN family. The epigenetic influence of ATF6 is present as seen in heterochromatin disorganization; however, the mechanistic role in this phenomenon was not elucidated. Considering the transcriptional regulatory role of ATF6 in ER stress response, additional genome-wide assays, such as chromatin immunoprecipitation (ChIP) and RT–PCR analysis, would help shed light on additional novel gene targets encoding direct epigenetic regulators.

Whereas MSCs are less prone to replicative stress, recent studies have found that age-associated replication stress-induced ER stress can cause accelerated cellular senescence, structural dysregulation of several organelles and, on the epigenetic level, heterochromatin in these cells (173). However, there are only a few studies that have reported replicative stress-associated epigenetic alterations, namely regarding histone accessibility and perhaps modifications (Table 1). Because of the lack of studies in this area, further studies are required to elucidate the mechanisms involved in the replicative stress-induced epigenetic alterations in MSCs and on other regulatory factors regarding all facets of epigenetics. This investigation would be of great interest for the current MSC therapies to address and resolve major challenges from cell isolation protocols to clinical application (174).

### ISC

ISCs are required to self-renew and differentiate at a high rate to regenerate the intestinal epithelium to maintain proper function and, thus, are subjected to replicative stress. This replicative stress has been linked to genome-wide alterations to the DNA methylome in highly proliferative tissues, including the intestinal epithelium, where methylation levels increased (114, 115, 175, 176). The ISC compartment is also subjected to hypermethylation as demonstrated in a computational model of ISC DNA repair showing the indirect influence of stress response on DNA methylation (177). In silico simulations displayed that promoter regions marked with H3K27me3 were found to undergo DNA hypermethylation during DNA repair in response to replicative stress in moderately irradiated ISCs (Fig 4A). This was associated with an increase in de novo DNA methyltransferase (DNMT) activity recruited to open chromatin regions undergoing repair and containing low levels of DNA methylation. The hypermethylation of gene promoters in open chromatin regions remained stable even after DNA repair, thus inducing long-term alterations to the epigenome. Consequences associated with this alteration include the development of aging phenotypes and tumorigenesis (178, 179).

Investigating the influence of replicative stress on the epigenetic landscape within ISCs is of importance because of its subjectivity to this intercellular stressor. The above studies have reported that replicative stress-induced epigenetic alterations and effects on regulators have been observed on the DNA and histone levels (Table 1), but further investigation on their mechanisms and downstream effects is needed. At the DNA level, hypermethylation

at promoter regions took place and remained stable even after DNA repair. This long-term alteration could allow for ailments such as cancer and age-associated phenotypes to develop over time. It is of interest to investigate stress response pathways such as those involved in DDR and tolerance that may induce an intracellular state (e.g., open chromatin) prone to de novo epigenetic modifications.

## Oxidative stress

In cells and tissues, an imbalance between the production and detoxification of ROS will induce oxidative stress. Free radicals and oxidants such as superoxide radicals, hydrogen peroxide, hydroxyl radicals, and diatomic oxygen are known ROS that upon accumulation will negatively affect several intracellular structures, such as organelle membranes, DNA, and protein. DNA lesions caused by oxidative stress are commonly because of the formation of ROS-induced 8-OHdG structures on DNA, which has demonstrated alterations within the epigenome resulting in overall genomic mutagenesis and consequential development of cancers (180, 181). In addition, oxidative stress has been shown to be associated with aging as it drives aging- and senescence-associated phenotypes in cells (182, 183). This has also been reported within several stem cell compartments under an aging, developmental, disease, and stress response context (184, 185, 186, 187, 188). These negative correlations have also been extended to epigenetic regulatory networks within several stem cell compartments.

### HSC
Oxidative stress has been causally linked to aging-associated phenotypes in HSCs, including an accumulation of DNA damage, apoptosis, telomere shortening, poor reconstitution capacity and loss of quiescence (91, 184, 189, 190). Within the last decade, there have been studies demonstrating the protective and maintenance role of conserved epigenetic regulators in HSCs, one of which involves PcG proteins.

PcG proteins are well known to be epigenetic regulators in gene silencing via histone modifications catalyzed by two major complexes, PRC1 and PRC2. PRC1 monoubiquitylates histone H2A at lysine 119 (H2AK119ub1), whereas PRC2 mono-, di-, and tri-methylates histone H3 at lysine 27 (H3K27me1/2/3) (191). Among the two PcG complexes, Bmi1, a subunit of PRC1, has been reported to have critical roles in maintaining the self-renewal and multipotent differentiating capacities in HSCs (192, 193, 194, 195) (Fig 2B). Collectively, it has been well established that Bmi1 is essential for the maintenance of normal HSC function under homeostatic conditions. In competition repopulation assays and in vitro studies, Bmi1 has been shown to confer protection against both replicative and oxidative stress in highly purified CD34⁻ LSK HSCs from *Tie2-Cre; R26StopFLBmi1* mice where *Bmi1* (92). In addition, purified CD34-LSK HSCs from *Tie2-Cre;R26StopFLBmi1* mice cultured in buthionine sulfoximine, for oxidative stress induction, showed significantly more colonies than control HSCs with buthionine sulfoximine. This suggests that Bmi1 can enhance and maintain the self-renewal capacity as part of its cytoprotective role against oxidative stress. Intracellular ROS levels were also measured and found to be similar in HSCs and downstream progenitors between those with

overexpressed *Bmi1* and the control. From this observation, Bmi1 could then be described to confer resistance to oxidative stress in HSCs not by regulating the production/accumulation of ROS, rather through some unknown mechanism that allows HSCs to tolerate high levels of ROS. This is an interesting contrast to another study reporting that Bmi1 is essential for regulating a set of genes critical for mitochondrial function and ROS production (196).

The mitochondria are a known origin of ROS as they facilitate oxidative metabolism. If left metabolically active without regulation, aging-associated phenotypes such as altered fate decisions and dysregulated self-renewal maintenance may occur in HSCs. This is especially detrimental to the HSC compartment as it is necessary to largely remain quiescent to retain lifelong functionality. Autophagy has been reported to have an active role in regulating oxidative stress through suppressing metabolism by clearing mitochondria in young and old HSCs from mice (197). When autophagy was repressed, qRT-PCR and gene ontology analysis in HSCs showed a significant increase in hypomethylation in regions expressing genes involved in fate decisions upon the activation of oxidative metabolism. This was associated with accelerated myeloid differentiation, a known aging hallmark in HSCs. Interestingly, when comparing populations of old HSCs, only a small subset can activate autophagy despite an overall competency for autophagy induction. In consideration of these findings, age may be associated with lessened resistance against oxidative stress in older HSCs, causing long-term alterations in the methylome that confers aging-associated phenotypes.

There are clear indications of altered epigenetic mechanisms both in response to and a result of oxidative stress, under certain conditions such as aging (Table 1). Bmi1, a subunit of the histone modifier PRC1, displayed cytoprotective roles in enhancing tolerance against oxidative stress in HSCs. Its mode of action pertains to the maintenance and enhancement of HSC self-renewal capacity; however, the mechanism behind this is unknown. Investigating this pathway and the role of PRC1 in modifying the ubiquitination landscape among histones would be is warranted. Conversely, when left to its own devices, oxidative stress is shown to induce hypomethylation in regions promoting myeloid differentiation, thus altering HSC fate decisions towards the myeloid lineage similar to changes in the aged HSCs compartment where there is an expansion of myeloid biased stem cells (101, 198). These studies have only unveiled replicative stress-induced epigenetic alterations on the DNA and histone levels and its regulators. Thus, it would be of interest to elucidate the molecular mechanisms and additional potential epigenetic regulators involved in these long-term insults to the epigenome during oxidative stress for the regenerative potential of BM transplantation. In addition, investigating the cytoprotective roles of epigenetic regulators in response to oxidative stress would identify targets for epigenetic therapies involving hematopoietic diseases, such as leukemia.

### MSC
MSCs have widely demonstrated its therapeutic potential in tissue repair and overall regenerative capacities in transplantation studies. However, transplantation and prolonged culturing of MSCs is associated with the induction of oxidative stress, resulting in MSC exhaustion and accelerated aging (102, 103).

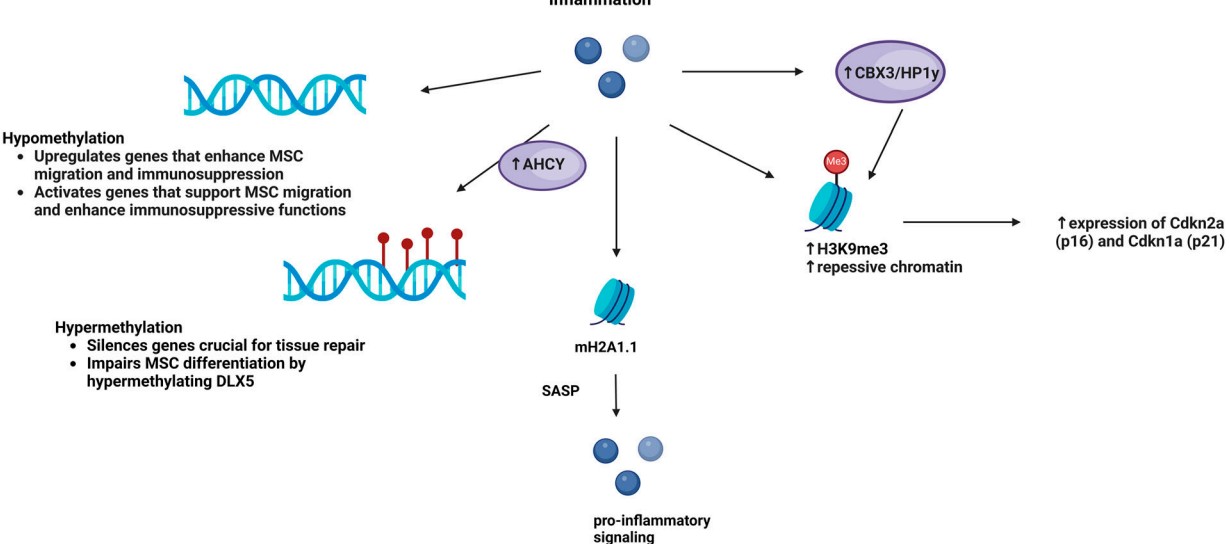

**Figure 3. A schematic illustration of epigenetic changes in MSCs because of various stressors.**
**(A)** In MSCs, ROS-induced DNA damage induced prolonged DDR activation and increased yH2AX foci driving premature senescence. In addition, miR-210, miR-29a-3p, and miR-30c-5p confer resistance to oxidative stress in MSCs, although upstream regulatory mechanisms of these miRNAs require further investigation. In aging MSCs, oxidative stress inhibits osteogenic differentiation by increasing EZH2 levels, which elevate H3K27me3 at the Foxo1 promoter. **(B)** Dnmt3b was found to bind to the *Shh* promoter region and directly catalyze hypermethylation. HDAC1 negatively correlates with osteogenesis by deacetylating the Jag1 promoter in BMSCs, down-regulating JAG1 and the NOTCH signaling pathway, whereas mechanical stimulation reduces HDAC1 levels, rescuing JAG1-mediated osteogenesis. **(C)** Histone variant mH2A1.1 modifications play a significant role in epigenetic regulation via the TLR4 pathway, inducing SASP. Increased H3K9me3 in aged MSCs leads to a repressive chromatin state, cellular senescence, reduced self-renewal capacity, and elevated expression of aging-related genes such as Cdkn2a (p16) and Cdkn1a (p21), with KDM4B loss exacerbating these effects by raising H3K9me3 and HP1α levels, ultimately impairing MSC functionality and regenerative capacity. Overexpression of AHCY leads to genome hypermethylation, whereas CBX3/HP1γ contributes to increased levels of H3K9me3.

Emerging evidence has reported the role of epigenetic regulations and alterations contributing to oxidative stress or mounting antioxidant defense mechanisms against it. In human endometrium-derived MSCs (hMESCs), prolonged DDR activation and increased yH2AX foci because of ROS-induced DNA damage were observed (103) (Fig 3A). It is the first study to report a molecular

mechanism driving premature senescence of hMESCs under oxidative stress. hMESCs exposed to sublethal oxidative stress exhibited premature senescence, which is suggested to be attributed to increased p38MAPK/MAPKAPK-2 pathway activation. The downstream effects of p38MAPK activation involved permanent ROS production, resulting in prolonged DDR signaling. DDR activation is known to induce phosphorylation of γH2AX in mammalian cells and tissues (199). It could then be said that oxidative stress is causally linked to γH2AX formation as a potential epigenetic histone modification, simply rather than a DNA damage marker. To address this notion, further investigation showing overlap of γH2AX as an epigenetic regulator of rDNA or novel transcriptional regulatory roles is needed.

microRNA-210 (miR-210) has been reported to confer resistance to oxidative stress in MSCs both in in vitro and in vivo settings. In functional assays on cultured MSCs isolated from the BM of Sprague-Dawley rats, overexpression of miR-210 resulted in a significant reduction in apoptosis and an increase in cell viability under oxidative stress in a concentration-dependent manner (200). This was accompanied by an increase in SOD activity and a decrease in malonaldehyde and ROS concentrations, all of which are reliable markers of oxidative stress (201, 202, 203). miR-210 was found to be a key activator in the c-Met pathway, of which it acts upstream of ROS generation. Under oxidative stress without miR-210 overexpression, c-Met was found to be repressed, resulting in increased levels of ROS. However, the exact mechanisms were not addressed. This suggests that ROS acts as a suppressor of the c-Met pathway, thus down-regulating downstream targets involved in clearing ROS. However, upon robust miR-210-mediated activation of c-Met, antioxidant defenses may be mounted. Both in vitro and in vivo studies using mice with acute liver failure and isolated human adipose-derived MSCs reported that under oxidative stress, miR-210 is naturally up-regulated from basal levels to maintain normal MSC function and clear cellular and mitochondrial ROS through ROS inhibitor iron-sulfur cluster assembly protein 1/2 (ISCU1/2) (104). Through treatment with the small molecule zeaxanthin dipalmitate (ZD), basal levels of miR-210 were increased and maintained at relatively high levels through inhibition of the PKC/MAPK/ERK pathway, which enhanced cell survival and resistance to oxidative stress in both in vitro and transplantation studies. Although pharmacological overexpression of miR-210 shows positive clinical application in enhancing MSC-based transplantation therapies, further investigation into the natural regulation of miR-210 under oxidative would be interesting.

Additional miRNAs, namely miR-29a-3p and miR-30c-5p, have also been shown to confer resistance and cytoprotection under oxidative stresses through computational and experimental approaches. Global depletion of miRNAs performed in hMSCs through silencing DiGeorge syndrome critical region 8 (DGCR8) gene, which is essential for miRNA biogenesis (105), resulted in impaired stem cell function, proliferation, premature senescence, and increased ROS production (204). Computational and in vitro analysis of antioxidant function and regulation revealed decreased SOD2 expression because of DNMT3A-mediated hypermethylation of upstream regulatory regions with respect to the SOD2 gene. Furthermore, miRNAs miR-29a-3p and miR-30c-5p demonstrated direct regulation of DNMT3A, to which overexpression of these

miRNAs rescued hMSCs from oxidative stress and premature senescence. This suggests a novel stress response pathway against damaging ROS levels involving the miR-29a-3p/miR-30c-5p/DNMT3A/SOD2 axis. The regulation of these miRNAs, however, was not addressed. Further investigation on the upstream regulatory mechanisms controlling miRNA levels would further establish and elucidate this novel oxidative stress response axis.

Downstream effects of oxidative stress were found to extend to mitochondrial DNA (mtDNA) on an epigenetic level. In young human heart MSCs under replicative and oxidative stress-induced premature senescence, mtDNA methylation increased considerably, whereas subsequent COX2 gene expression decreased (205). Cytochrome c oxidase (COX) (also known as complex IV) catalyzes the reduction in oxygen to water in the electron transport chain. COX2 is one of three core mtDNA-encoded subunits forming complex IV (106). Given its enzymatic function, decreased expression of COX2 would induce mitochondrial dysregulation and allow for diatomic ROS accumulation. The mechanism in which COX2 mtDNA methylation occurs was partially elucidated through the treatment of 5-aza-2′-deoxycytidine (AdC), an inhibitor of DNMT1. Upon DNMT1 inhibition, COX2 mtDNA methylation decreased with subsequent increased protein expression. This was not surprising considering that DNMT1 contains a mitochondrial targeting sequence (206). The exact mechanistic function of DNMT1 in mtDNA methylation was not addressed nor was the association of the specific methylated CpG COX2 sites to COX2 suppression and mode of regulation. A functional analysis demonstrated a causal link between oxidative (and replicative) stress to epigenetic alterations taking place on the mtDNA methylome, specifically to the COX2 gene. Further investigations on the molecular mechanisms and associations of key epigenetic regulators, such as DNMT1, to mtDNA under stressed conditions are warranted.

On the flipside of downstream epigenetic alterations and responses to stress, upstream phenomena are also critical in elucidating the crosstalk between the epigenome and stress. In aging BM-derived MSCs, oxidative stress inhibits osteogenic differentiation through Wnt gene suppression in a mouse model of osteoporosis facilitated by increased levels of EZH2, a conserved methyltransferase of histone H3K27 trimethylation (207). Aging MSCs that also displayed EZH2 overexpression had higher levels of ROS and conversely displayed a significant reduction because of EZH2 knockdown. Through CHIP and RT–PCR analysis, EZH2 was shown to increase H3K27me3 levels in the promoter region of Foxo1 and was supported by lower expression levels. FOXO1 is a critical transcriptional regulator of antioxidant agents such as CAT and SOD2. Thus, EZH2 elicits oxidative stress by allowing ROS levels to increase as a result of down-regulating FOXO1-mediated antioxidant agents. EZH2 perhaps could become an effective pharmacological target for epigenetic therapy in reversing oxidative stress induced by varying factors outside of aging.

In clinical settings, MSCs are highly regarded for their therapeutic potential as previously mentioned. However, protocols in the expansion and usage of MSCs need further optimization because of their being prone to oxidative stress (102, 103). The current studies show potential means for improving MSC survivability and viability via ROS reduction, namely through miRNA induction as reported above (Table 1). It would be of interest to further investigate the

mechanism in the biogenesis and downstream effects of these miRNAs in MSCs undergoing oxidative stress. In addition, functional and transplantation assays involving external induction of these miRNAs in primary MSCs would be critical for gauging its efficacy and practical usage. In addition, external inhibition of epigenetic regulators promoting ROS production such as EZH2 (207), could prove to be an effective means of reducing oxidative stress when expanding and transplanting MSCs in a clinical setting (Table 1). These studies warrant further investigation into these epigenetic factors and identifying others.

### NSC

Adult neurogenesis, which is essential for brain function, is characterized by a high rate of energy consumption and thus leads to the accumulation of ROS. Effective clearance of ROS is widely recognized as one of the critical factors in promoting brain rejuvenation and preventing disease progression (208). Conversely, accumulation of ROS promotes DNA damage and cell death, which may contribute to the pathogenesis of neurodegenerative diseases such as AD (88, 209).

One of the causes of significant increases of ROS production in NSCs is mitochondrial dysfunction. It has been suggested that this dysfunction, which results in increased ROS and impaired mitochondrial respiration, underlies defects in adult neurogenesis during aging (116). Some have suggested that antioxidant compounds (e.g., docosahexaenoic acid) may abate the adverse effects of ROS and, therefore, improve adult neurogenesis and delay the onset of neurodegeneration (208, 209). Oxidative stress sensing also has a role in regulating the balance between self-renewal and differentiation of NSCs. Namely, whereas increased ROS levels are typically associated with NSC proliferation and differentiation, recent findings showed that quiescent NSCs in the hippocampus exhibited the highest levels of ROS, adding a level of complexity to the relationship between redox regulation and NSC fate decision (210). Oxidative stress also up-regulates SIRT1, which can suppress NSC self-renewal and shift their fate towards the astroglial lineage (210). To improve the cognitive healthspan, it would be interesting to examine how compounds that are explored in the geroscience context, such as spermidine and fisetin, and which reduce oxidative stress and neuroinflammation, act specifically on NSCs.

### ISC

Studies investigating the epigenetic response or modulations in ISCs under oxidative stress are scarce. This itself warrants further investigation considering the causal and protective roles of several epigenetic regulators seen in other stem cell compartments. In support of this, Park et al were the first to show direct evidence of age- and oxidative stress-induced accumulation of DNA damage in ISCs, alongside the use of yH2AvD, analogous to mammalian yH2AX, as a biomarker (211) (Table 1) (Fig 4B). Several studies have shown yH2AX as a short- and long-term epigenetic histone modification recruiting chromatin remodeling proteins for increasing chromatin accessibility during DNA damage repair (212, 213, 214). As mentioned previously, in old cycling HSCs, yH2AX was reported as a long-term histone modification suppressing rDNA and subsequent ribosome biogenesis (29). Collectively, yH2AX, and perhaps its analog yH2AvD, displays an epigenetic role largely under stress-induced DNA damage in somatic cells, including stem cells. Further investigations to elucidate this notion would be needed.

### MuSC

Epigenetic regulation of MuSCs is required to maintain stem cell homeostasis and myogenic differentiation potential for skeletal muscle repair. Part of this process, and during endurance exercise, involves a slight increase in skeletal muscle ROS production for ensuring healthy mitochondrial function and adaptations (215). However, excess ROS production may still arise and induce oxidative stress (119). Bmi1 is shown to maintain MuSC pools and myogenesis during muscle via repression of the *ink4a* locus (216, 217) after muscle injury and during oxidative stress. In addition, Bmi1 overexpression in mouse models of dystrophinopathies and DMD patients induced enhanced metallothionein 1 (MT1)-mediated protection against oxidative stress, which improved muscle strength and regeneration (218, 219). Functional analysis on MT1-mediated oxidative stress displayed increased ROS and 8-OHdG in cultured MuSCs of MT1 knockdown mice when compared with controls. The mechanism underlying the observed MT1-mediated protection against oxidative stress has not been addressed and warrants further investigation. Bmi1 is involved in gene silencing through histone modifications as an integral component of PRC1, which suggests that MT1 enhancement is a result of repressing a MT1 repressor (Fig 6A). In MuSCs of DMD patients and quiescent cells, Bmi1 and MT1 expression levels were reduced both at the protein and RNA levels (219). Increased Bmi1 was associated with an enhanced energetic state, marked by increased ATP production during in vitro and in vivo studies of human DMD MuSCs (219). Interestingly, ROS generation did not increase, possibly because of MT1- and PRDX2-mediated antioxidant activities. PRDX2 is an antioxidant enzyme that scavenges cellular peroxides and facilitates redox homeostasis (220). Bmi1 confers resistance to oxidative stress in MuSCs, perhaps through gene silencing of repressors and/or transcription factors contributing to increased mitochondrial ROS generation.

As a result of cellular maintenance and myogenic differentiation, ROS levels in MuSCs are allowed to increase without passing a certain threshold that would otherwise induce oxidative stress. In response to increasing ROS levels, the Bmi1 has shown to facilitate histone modifications which in turn mounts antioxidative mechanisms conferring resistance to oxidative stress (Table 1). However, its direct role and mechanism has yet to be elucidated. This warrants further investigation into the role of histone modifiers, and other epigenetic regulators, in response to rising ROS levels and effects on antioxidative mechanisms.

## Mechanical stress

Mechanical stress can be defined by deformation of dynamic cellular structures, such as the cytoskeleton, because of mechanical forces and its resistance to deformation (221). The mechanical force applied to cells elicits a physical stimulation or activation of cell surface receptors (such as integrins and cadherins) that either mediate communication with the ECM or adjacent cells.

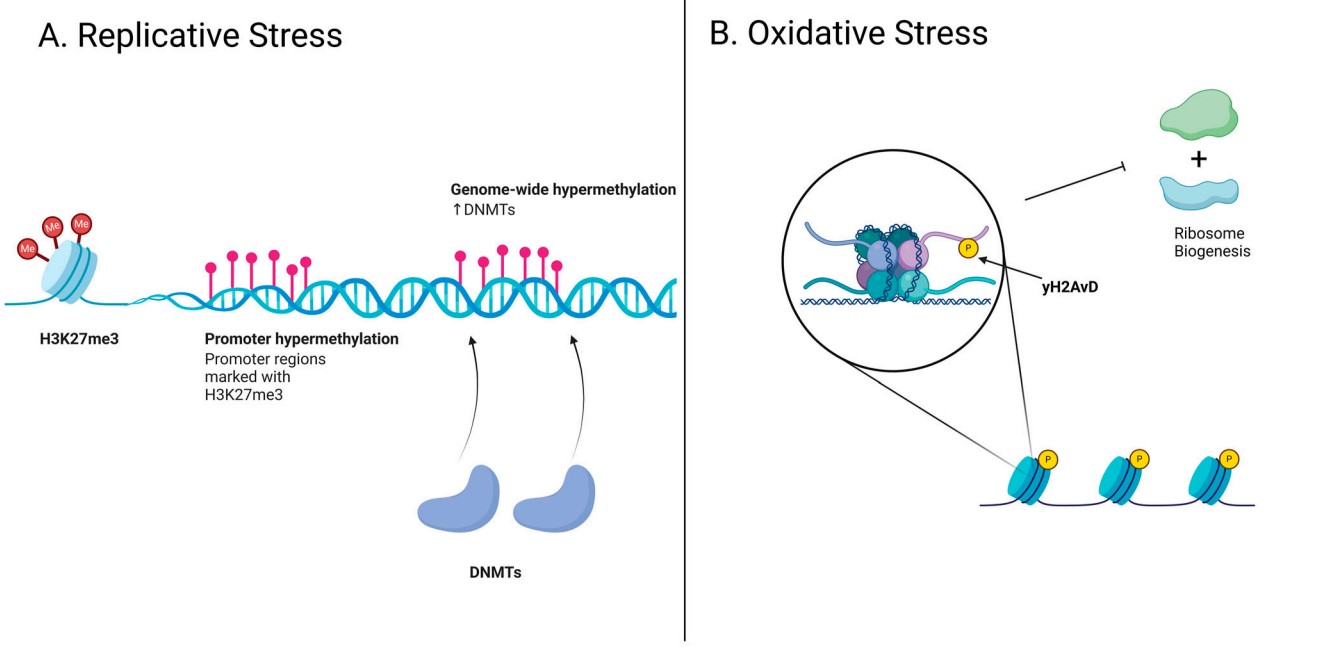

## A. Replicative Stress

**Genome-wide hypermethylation**
↑DNMTs

**H3K27me3**

**Promoter hypermethylation**
Promoter regions
marked with
H3K27me3

**DNMTs**

## B. Oxidative Stress

+
Ribosome
Biogenesis

yH2AvD

## C. Inflammatory Stress

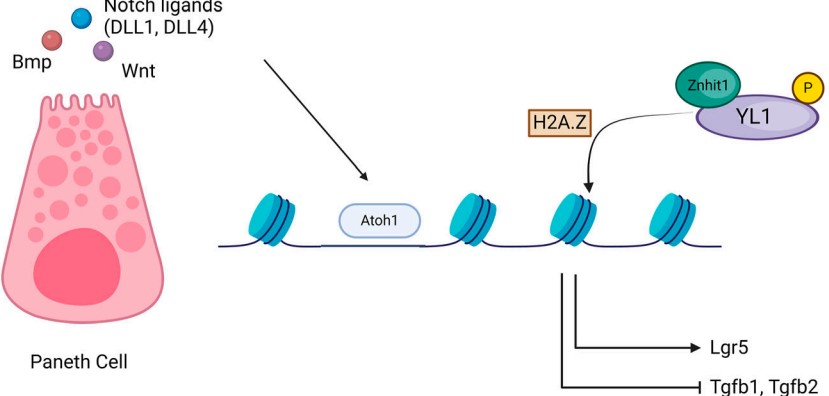

Notch ligands
(DLL1, DLL4)

Bmp

Wnt

Znhit1

P

YL1

H2A.Z

Atoh1

Paneth Cell

Lgr5

Tgfb1, Tgfb2

**Figure 4. A schematic illustration of epigenetic changes in ISCs because of various stressors.**
**(A)** Replicative stress has been linked to genome-wide hypermethylation to the DNA methylome in the intestinal epithelium. Simulations displayed that promoter regions marked with H3K27me3 were found to undergo DNA hypermethylation during DNA repair in response to replicative stress in irradiated ISCs. This was associated with an increase in DNMT activity recruited to open chromatin regions undergoing repair and containing low levels of DNA methylation. **(B)** ISCs under oxidative stress demonstrate age- and stress-induced DNA damage accumulation in ISCs using yH2AvD which suppresses ribosome biogenesis. **(C)** Signaling molecules such as Wnt, Bmp, and Notch, produced by Paneth cells, regulate ISC fate and function by allowing Atoh1 transcription factor to bind, enabling cells to revert to a primitive state. Epigenetic modifications such as the incorporation of histone variant H2A.Z mediated by Znhit1, are central to ISC plasticity, influencing key genes such as Lgr5 and TGF-$\beta$ crucial for self-renewal and differentiation, with specific roles for MTG8, MTG16, and Id3 in modulating chromatin accessibility and maintaining ISC identity.

### *MSC*

Skeletal loading (increased mechanical stress on bone) has been shown to be essential in regulating BM MSCs (BMSCs) for osteogenic differentiation during bone repair, development, and mineralization (107, 222, 223, 224). Epigenetic modifications involving DNA methylation and histone modifications are reported to regulate osteogenesis in response to mechanical stimulation in different manners.

In vitro studies on mouse MSCs cultured in media conditioned from mouse osteocyte cultures shortly exposed (24 h) to fluid shear stress (a proxy for mechanical stress) displayed a decrease in DNA methylation in adipogenic and osteogenic markers by at least 25% and 30%, respectively (225). Although genes within these regions become more accessible, transcriptomic analysis revealed increased expression in only late osteogenic markers and little to no

change in early osteogenic markers and overall adipogenic markers. DNA methylation decreased to a higher degree in late osteogenic markers, allowing for its expression. It is possible that downstream pathways of mechanical stress prime MSCs for osteogenic differentiation through expression of late markers and increased gene accessibility in early markers via decreased DNA methylation. The connection between mechanical stress and direct effects on epigenetic modifications is intriguing, and it will be interesting to better understand these relationships.

Additional osteogenic-promoting DNA regions in BMSCs also underwent hypomethylation upon mechanical stimulation as shown in in vivo and in vitro studies (226). In BMSCs of young mice, mechanical stimulation via cyclic mechanical stretch was shown to activate the Hedgehog (Hh) signaling pathway, demonstrating a role as an endogenous mediator for promoting osteogenesis. Under normal conditions, methylation of the promoter and enhancing regions of *Sonic hedgehog* (*Shh*), a direct upstream activator of the Hh signaling pathway, was present, resulting in gene silencing (227, 228). Through ChIP analysis, Dnmt3b was found to bind to the *Shh* promoter region and directly catalyze hypermethylation. However, mechanical stimuli on BMSCs demonstrated direct down-regulation of Dnmt3b protein levels accompanied by downstream up-regulation of *Shh* and subsequent activation of Shh-mediated osteogenesis. Regarding *Shh* demethylation, no mechanism nor epigenetic regulator was addressed. Ten-eleven translocation (TET) enzymes are conserved dioxygenases that mediate DNA demethylation and have been reported to regulate osteogenic differentiation in MSCs by increasing chromatin accessibility (229). TET-mediated demethylation possibly has a direct role in *Shh* up-regulation by reversing Dnmt3b-mediated methylation.

Histone deacetylases (HDACs) are negatively correlated with osteogenesis, namely with the expression of osteogenic markers in MSCs (112, 230). In BMSCs of mice and healthy human donors, HDAC1-mediated histone H3 deacetylation of the *Jag1* promoter was present, resulting in down-regulation of JAG1 (231). JAG1 is a direct activator of the NOTCH signaling pathway and is essential for the induction of osteogenic differentiation in BMSCs (232, 233). The downstream effects of JAG1 down-regulation included the repression of the NOTCH signaling pathway and ultimately inhibited osteogenesis (Fig 3B). Mechanical stimulation via cyclic mechanical stretching enhanced osteogenesis, even in mice that underwent skeletal unloading. ChIP analysis revealed that mechanical stimuli directly down-regulated HDAC1 protein levels, which in turn rescued JAG1-mediated osteogenesis. This was supported by increased levels of osteogenic markers including ALP, COL1a1, and OCN.

BMSCs have demonstrated they maintain stem cell identity through epigenetic repression via DNA methylation and/or histone H3 deacetylation of promoter regions belonging to osteogenic differentiation-promoting genes (Table 1). Upon mechanical stimulation, the reversal of these epigenetic modifications is elicited through direct regulation of their respective modifiers, which is critical for bone development and formation. As such, hypomethylation and acetylation acts as an "ON" switch for osteogenesis in BMSCs upon mechanical stress whereas hypermethylation and deacetylation act as an "OFF" switch. In the current literature, mechanical stress-induced hypomethylation and acetylation were reported to have conferred osteogenic priming in MSCs (Table 1). It

would be of interest to further investigate the mechanism involved in the interplay between these epigenetic alterations for developing interventions against bone-weakening ailments (e.g., osteoporosis). These epigenetic regulators also show potential as targets for enhancing BMSC-based transplantation or direct pharmacological manipulation.

### MuSC

Skeletal muscle loading (increased mechanical stress on skeletal muscle tissue) activates the regulatory networks governing myogenic differentiation of MuSCs, which in turn can induce an anabolic hypertrophic response, enhanced tissue function, and overall muscle regeneration (234, 235). Epigenetic regulators involved in maintaining stem cell identity or promoting myogenic differentiation have been reported to be differentially expressed upon exposure to mechanical stress.

In vitro and in vivo analysis using adult mice revealed protein arginine methyltransferases 5 and 7 (PRMT5/7) to promote MuSC self-renewal and myogenic differentiation during muscle regeneration (120, 121). Arguably, mechanical stress may be strongly connected in consideration that strenuous mechanical activities such as weightlifting are well-established inducers of muscle regeneration (122). PRMT5/7 catalyzes the transfer of methyl groups onto arginine residues of proteins, including histones (123, 124, 236, 237). The inactivation or KO of PRMT5/7 in adult MuSCs underwent cell-cycle arrest and premature senescence, which in turn inhibited muscle regeneration. CHiP analysis revealed a direct role of PRMT5 in repressing *p21* expression via histone dimethylation of H3R8 (H3R8me2) in two of four regulatory sites upstream of the murine *p21* gene: upstream enhancer-like region (En) and p53 binding site (p53BS). However, inactivation of p21 in MuSCs of *Prmt5 KO* mice resulted only in increased MuSC proliferation and number of myogenic colonies, but it did not fully restore muscle regenerative capacity. This suggests that PRMT5 is involved in an epigenetic regulatory network governing several facets of MuSC function and maintenance outside of *p21* regulation. Differing from PRMT5, PRMT7 regulates *p21* expression via DNMT3b-mediated hypermethylation of two CpGs at the *p21* promoter (Fig 6B). In addition, *Dnmt3b* and *p21* promoter regions exhibited decreased H4R3me2 levels and a decrease and increase in the activation mark H3K4me3, respectively. However, a CHiP analysis of PRMT7 at each promoter site was not performed, which warrants further investigation into a possible direct regulatory role for *p21* expression.

The epigenetic regulatory roles of PRMT5 and 7 in promoting muscle regeneration displayed a contrasting interplay between histone methylation sites of tumor suppressor genes *p21* and *p53* and *Dnmt3b* promoter regions (Table 1). In addition, PRMT7 facilitated DNMT3b-mediated DNA hypermethylation of the *p21* promoter region. The epigenetic mechanisms displayed during muscle regeneration were not addressed under mechanically stressed conditions However, it is arguable that these epigenetic alterations behave similarly, if not the same, in MuSCs under direct mechanical stress as muscle regeneration takes place after mechanically stressed muscle after strenuous exercise and muscle tissue hypertrophy (122). It would be intriguing to perform additional studies confirming this notion. The effects of mechanical stress on MuSCs have been widely studied (238); however, studies regarding its

influence on the epigenetic landscape are scarce and would be of interest to further investigate.

## Inflammatory stress

Inflammation is a crucial immune response designed to protect the body from adverse stimuli, such as injury and pathogens. It is characterized by the activation of immune cells, release of cytokines, and increased blood flow to affected areas. During acute inflammation, this defense system works to remove these damaging agents and begin the process of restoring tissue function. Failure to eradicate factors causing the acute inflammation can lead to chronic inflammation if left unchecked. Chronic inflammatory diseases involve consistent inflammation that can damage tissues and weaken tissue function, further driving inflammation (239). Inflammation is seen to be prevalent in aging tissues, as gene analysis between young and aged tissues shows up-regulation of genes associated with inflammation (240). In addition, chronic inflammation expedites the aging process of immune cells, leading to compromised immune function and inability to clear senescent cells.

### HSC

Inflammation serves as a key trigger for activating dormant HSCs during infections or other hematological stresses (241). Immature HSCs typically in a quiescent state, can be prompted to proliferate by inflammatory stimuli, acting as a reserve directed to produce mature blood cells during stress (242, 243). However, prolonged chronic or acute inflammatory stress can debilitate HSCs, leading to reduced regenerative ability and a bias towards myeloid cell differentiation, thereby impacting long-term function (244).

HSCs are critically influenced by inflammation and infections, impacting their functionality, differentiation, and long-term viability. Acute inflammation is typically a transient response aimed at restoring tissue homeostasis, but chronic inflammation can disrupt HSC homeostasis. Infections and persistent inflammatory signals can lead to an overproduction of myeloid cells at the expense of lymphoid cells, which skews the differentiation of HSCs. This myeloid bias, often observed in aged individuals, is driven by pro-inflammatory cytokines like IL-1, IL-6, TNF-α, IFN-α, and IFN-γ (93, 126, 245, 246, 247, 248). These cytokines can disrupt the BM niche by altering the interactions between HSCs and their supportive microenvironment. For instance, IFN-γ can displace HSCs from quiescence-promoting CAR cells via increased BST2 expression (95). This relocalization reduces the influence of quiescence maintaining signals from CAR cells, prompting HSC activation and increased differentiation. The BST2 interaction enhances HSC binding to E-selectin, facilitating movement within the BM and subsequently more prone to activation and depletion. In addition, IL-1 significantly accelerates HSC differentiation, as demonstrated by the faster division rates and enhances expansions of HSCs treated with IL-1 (248). This effect is mediated through early activation of the transcription factor PU.1, which drives myeloid lineage commitment. The study goes further to show that, chronic exposure to IL-1 results in a sustained increase in myeloid cell production at the expense of other lineages but compromises the long-term regenerative capacity of HSCs. Continuous activation through

inflammatory signals can exhaust HSCs, reducing their ability to stay quiescent, self-renew, and maintain long-term hematopoiesis.

Specific epigenetic regulators are affected by inflammatory signals, exacerbating the aging and functional decline of HSCs. The histone demethylase UTX/KDM6A, which maintains HSC potential decreases in expression with age and inflammation. This reduction leads to an aged gene expression profile in HSCs, characterized by impaired DNA repair mechanisms and increased oxidative stress (94). Mice with deficient UTX exhibited trilineage dysplasia, a condition such as myelodysplastic syndrome (MDS) in humans. Inflammatory cytokines can also influence the activity of enzymes such as DNMT3A and TET2, which are crucial for DNA methylation and demethylation processes. For example, elevated levels of TNF-α enhances the proliferative advantage of TET2 deficient HSCs (249), and IFN-γ can drive expansion of DNMT3A knockout cells (250). Mutations in these genes are commonly found in clonal hematopoiesis and are associated with an increased risk of hematological malignancies. Inflammatory conditions can further promote the expansion of these mutated HSC clones, increasing the likelihood of disease progression.

One notable change is the increased expression of P-selection, a cell surface adhesion molecule associated with physiological stress, including inflammation and aging (251). In HSCs, there is a substantial rise in P-selectin (*Selp* or *CD62P*) levels on the surface of aged and inflamed HSCs (98, 252). With age, HSCs exhibit increased expression of P-selectin, which typically is exclusive to activated platelets and endothelial cells during an inflammatory response (99). This suggests that HSCs are exposed to a heightened inflammatory state within the BM. The NF-κB pathway, a critical mediator of inflammation and *selp* expression, is significantly activated in aged HSCs, as evidenced by enhanced nuclear localization of the p65 subunit older cells (253). This activation results in the up-regulation of several inflammation-related genes, which further perpetuates the inflammatory state and contributes to overall functional decline in HSCs (Fig 2C). Another striking example is the inappropriate expression of the *IgK* gene in aged HSCs, which is not normally active in these cells but becomes highly expressed because of a loss of epigenetic regulation at the *IgK* locus. Furthermore, global analyses of gene expression have identified significant down-regulation of genes involved in chromatin remodeling and transcriptional silencing, such as the SWI/SNF-related chromatin remodeling genes (253, 254). This lack of regulation suggests a broader loss of transcriptional control across the genome, leading to inappropriate gene activation and increased risk of genomic instability.

Lastly, chronic inflammation affects HSCs through the induction of a senescence-associated secretory phenotype. Senescent HSCs secrete pro-inflammatory cytokines, growth factors, and proteases that create a feedback loop, perpetuating the inflammatory state and further impairing HSC function (255). This senescence-associated secretory phenotype contributes to the chronic inflammatory environment in the BM niche, exacerbating the decline in HSC function and promoting the development of hematopoietic disorders.

On the other hand, transient immune challenges can imprint long-term functional modifications. Acute LPS exposure triggers rapid, temporary changes in HSC proliferation and gene expression,

which normalize within days (110). However, LPS also induces persistent alterations in chromatin accessibility, specifically in myeloid lineage enhancers, enhancing the responsiveness of genes involved in immunity. The transcription factor C/EBPβ is crucial for maintaining these LPS-induced epigenetic marks and gene expression changes. The deletion of C/EBPβ eliminated the long-term epigenetic marks and gene expression changes induced by LPS, highlighting its role as a pioneer factor that initiates and sustains chromatin accessibility. Direct TLR4 signaling in HSCs is essential for these modifications, as TLR4-deficient HSCs show significantly reduced chromatin changes and responsiveness. This strategic adaptation enables a rapid and robust immune response upon reinfection. In another study, findings reveal that such impairments can be irreversible, with no recovery observed even after a year (108). Exposure to inflammation or bacterial infection leads to a permanent depletion of functional HSCs, resulting in accelerated aging and the emergence of blood and BM phenotypes typically seen in elderly humans but not in aged laboratory mice. During and after inflammatory challenges, HSCs fail to undergo self-renewal, highlighting a cumulative inhibitory effect of discrete inflammatory events over time. This research positions early and mid-life inflammation as a key mediator of lifelong defects in tissue maintenance and regeneration, significantly impacting HSC functionality and potentially leading to age-associated hematologic diseases. Early inflammatory responses inscribe a lasting memory in HSPCs, which can enhance the host's innate immunity against future infections.

### MSC

MSCs are profoundly influenced by inflammatory environments, leading to significant epigenetic modifications that impact their functionality and therapeutic potential. Inflammatory cytokines, such as TNF-α and IL-1β, can induce changes in DNA methylation and histone modification, thereby altering the expression of genes involved in MSC differentiation, proliferation, and migration. These epigenetic changes can enhance or impair the regenerative capacity of MSCs. For instance, inflammation-induced hypomethylation can up-regulate genes that enhance MSC migration and immunosuppression, whereas hypermethylation may silence genes crucial for tissue repair. hMSCs conditioned with Skov-3 medium for 16 d exhibited high levels of tumor associated fibroblasts markers Tn-C, TSP, and FSP and increased expression of a-SMA, FAP, and desmin, which are associated with cell migration and tissue remodeling (111). The same study shows that MSCs contribute to the tumor microenvironment by secretive immunosuppressive and tumor-promoting factors, HFG, EGF, and IL-6, indicating that inflammation-induced hypomethylation can activate genes that support MSC migration and enhance immunosuppressive functions, aiding in tumor progression. Alternatively, the presence of mice MSCs in tumor microenvironment was associated with enhanced fibrotic response, suggesting hypermethylation of genes that are crucial for normal repair, leading to fibrosis as opposed to normal healing (109).

In addition to DNA methylation, histone modifications also play a significant role in the epigenetic regulation of MSCs under inflammatory conditions. In MDS, MSCs in the BM are reprogrammed to support disease progression. In these MDS-MSCs, histone variant isoform mH2A1.1 up-regulation drives pro-inflammatory signaling via the TLR4 pathway, inducing a senescence-associated secretory phenotype and alters the proteomic and metabolic profiles of MSCs (256) (Fig 3C). Further proteomic analysis has shown up-regulation of several proteins, including AHCY and CBX3/HP1γ. Overexpression of AHCY, a protein part of the DNMT1 interactome, leads to genome hypermethylation, whereas CBX3/HP1γ contributes to increased levels of H3K9me3.

Increased H3K9me3 in aged MSCs is associated with several detrimental effects on their functionality and regenerative capacity. Removing KDM4B leads to increased MSC aging and senescence, indicating that higher H3K9me3 levels contribute to this. Losing KDM4B significantly raises the levels of H3K9me3 and HP1α clusters as MSCs age, promoting a more repressive chromatin state (257). Elevated H3K9me3 also significantly increases the nuclear size of MSCs after multiple cell divisions and leads to higher expression of aging-related genes such as *Cdkn2a* (p16) and *Cdkn1a* (p21). Overall, the accumulation of H3K9me3 in aged MSCs is linked to increased cellular senescence, reduced self-renewal capacity, and the formation of senescence-associated heterochromatin foci, contributing to the decline in their regenerative potential and functionality.

During aging, MSCs experience significant epigenetic changes driven by chronic inflammation, exacerbating cellular senescence. Inflammatory cytokines such as TNF-α and IL-6 activate the NF-κB pathway, perpetuating a pro-inflammatory environment through ROS (125). This chronic inflammation leads to DNA methylation changes in genes regulating the cell cycle, DNA repair, and differentiation. For instance, hypermethylation of the *Dlx5* impairs MSC differentiation. Senescent MSCs show altered gene expression, including miRNAs, resulting in reduced proliferation and structural changes. Oxidative stress further drives these epigenetic alterations, increasing susceptibility to malignant transformation.

### ISC

ISCs play a crucial role in maintaining the homeostasis and regeneration of the intestinal epithelium, but their function and regulation can be significantly altered by inflammatory processes, particularly through epigenetic modifications. Key proteins such as IFN-γ and STAT1 are pivotal in mediating these effects. IFN-γ-STAT1 signaling pathway becomes hyperactive during inflammation, leading to up-regulation of MHC-II in aged ISCs (258). The heightened STAT1 transcription factor signaling leads to increased surface expression of MHC-II in aged ISCs. This interaction facilitates CD4+ T-cell engagement, perpetuating a cycle of inflammation and immune response.

Epigenetically, inflammation induces significant changes in chromatin accessibility within ISCs. These modifications were examined using single-cell RNA sequencing and gene set enrichment analysis to identify differentially expressed genes and altered pathways. A study revealed that aged ISCs exhibit specific changes in chromatin that remain stable even when cultured in organoids ex vivo (258). These changes include increased accessibility at inflammation-associated loci, which promote a state of sustained inflammation within the epithelium. The differential gene expression analysis identified significant DEGs across

# Inflammatory Stress

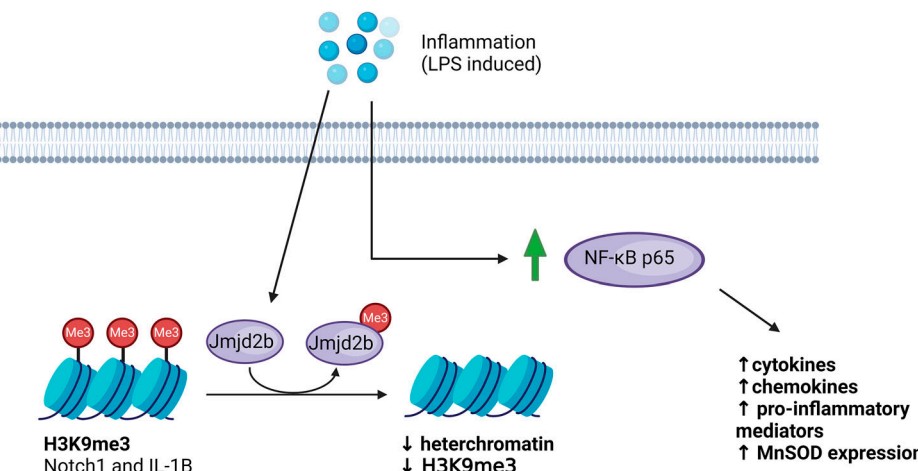

**Figure 5. A schematic illustration of epigenetic changes in NSCs because of stress.**
LPS adversely affects NSC viability and differentiation by inducing key epigenetic changes, particularly histone modifications, with prolonged LPS exposure increasing Jmjd2b and NF-κB p65 expression, leading to reduced H3K9me3 levels at gene promoters essential for NSC function and neurogenesis, thus profoundly altering the epigenetic landscape of NSCs.

various cell types in the intestinal epithelium, indicating that inflammation affects not only ISCs but also TA cells and enterocyte progenitors.

Inflammation triggers a complex array of epigenetic modifications in ISCs, influencing their function and regenerative capacity. Key proteins such as Hopx and Atoh1 are involved in these modifications, playing significant roles in the regulation of ISC differentiation and maintenance (142, 259). During chronic inflammation, such as in inflammatory bowel disease, ISCs are exposed to persistent inflammatory signals that can lead to lasting changes in their gene expression profiles. Inflammatory cytokines, such as IL-11 and type I interferons, have been shown to influence ISC behavior. IL-11, for example, is necessary for mucosal regeneration, whereas type I interferons can impair recovery and reduce ISC proliferation. These inflammatory signals can induce epigenetic modifications, such as DNA methylation, histone modifications, and changes in chromatin structure. For instance, prolonged ER stress and hypoxia, common in chronic inflammation, activate the unfolded protein response, which is crucial for ISC proliferation and differentiation. Studies have shown that HNF4α, a transcription factor involved in ER stress response, is essential for intestinal epithelial repair, linking inflammation-induced ER stress to epigenetic regulation of ISC function (260, 261).

Epigenetic regulation because of inflammation can result in the activation or repression of specific genes that are critical for ISC function. For example, studies using ChIP have demonstrated that inflammatory cytokines can lead to increased histone acetylation at loci associated with pro-inflammatory genes, enhancing their expression. Conversely, DNA methylation analysis has shown that inflammation can cause hypermethylation of genes involved in maintaining stem cell pluripotency, thereby reducing their expression and impairing the regenerative capacity of ISCs. In addition, single-cell RNA sequencing of ISCs isolated from inflamed tissues has revealed altered expression patterns of key regulatory

genes, highlighting the impact of chronic inflammation on the epigenetic landscape of these cells.

Recent evidence suggests that not only TA cells but also fully differentiated intestinal epithelial cells possess the potential to de-differentiate and replenish the damaged ISC niche. This raises critical questions about the inherent fate of these cells and the epigenetic mechanisms underlying their differentiation. Signaling molecules such as Wnt, Bmp, and Notch, produced by Paneth cells and myofibroblasts, play a pivotal role in regulating ISC fate and function (260). These pathways interact with the chromatin landscape, which, when in a permissive state, facilitates the binding of lineage-defining transcription factors such as Atoh1 (262). This chromatin accessibility allows for dynamic responses to injury, enabling differentiated cells to revert to a more primitive state and contribute to epithelial repair. Such plasticity is critical in maintaining intestinal homeostasis, particularly in response to acute injuries where the rapid regeneration of the epithelial barrier is essential.

Epigenetic modifications, particularly those involving chromatin structure, are central to the regulation of ISC plasticity. Histone variants, such as H2A.Z, play a significant role in remodeling chromatin and altering gene expression (263). The incorporation of H2A.Z into specific genomic regions, mediated by components such as *Znhit1*, influences the expression of key genes such as *Lgr5* and *TGF-β*, which are crucial for ISC self-renewal and differentiation (264) (Fig 4C). Deleting ZNHIT1 impacts ISC differentiation directly, without altering Wnt signaling, underscoring its specific role in regulating chromatin dynamics and gene expression in ISCs. Moreover, the transcriptional co-repressors MTG8 and MTG16, and factors such as Id3, modulate chromatin accessibility, maintaining ISC identity and controlling differentiation into secretory lineages (265). The interplay between these epigenetic regulators and transcription factors ensures a balanced differentiation process, allowing for efficient repair and regeneration of the intestinal epithelium.

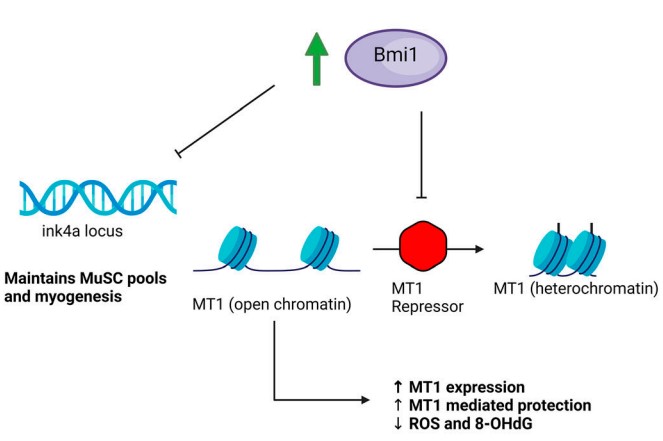

## A. Oxidative Stress

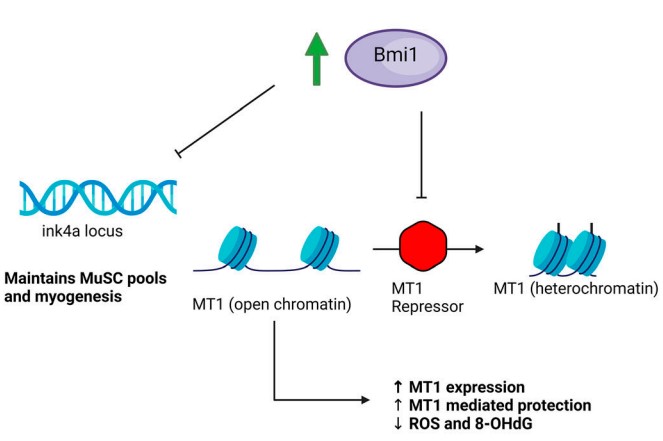

## B. Mechanical Stress

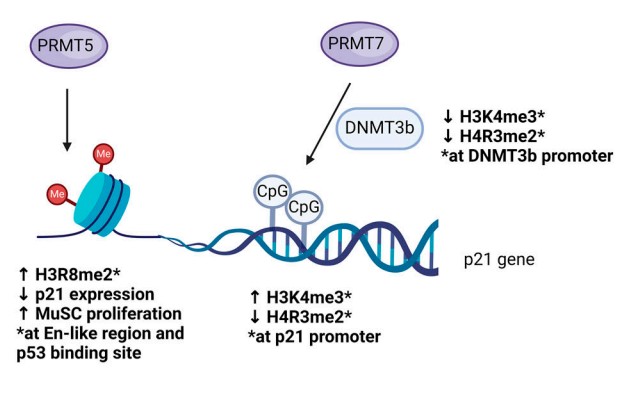

## C. Inflammatory Stress

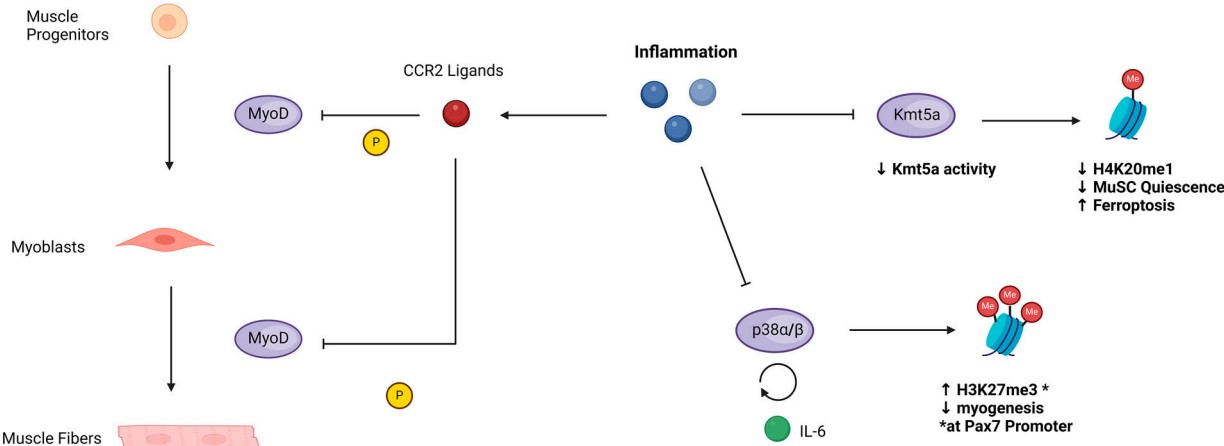

**Figure 6. A schematic illustration of epigenetic changes in MuSCs because of various stressors.**
**(A)** Bmi1 is shown to maintain MuSC pools and myogenesis during muscle via repression of the *ink4a* locus after muscle injury and during oxidative stress. Bmi1 overexpression in mouse models and patients enhances MT1-mediated protection against oxidative stress, improving muscle strength and regeneration, likely through gene silencing mechanisms involving PRC1. **(B)** PRMT5 represses p21 expression via H3R8 dimethylation at regulatory sites upstream of the murine p21 gene. PRMT7 regulates p21 expression through DNMT3b-mediated hypermethylation at the p21 promoter, influencing H4R3me2 and H3K4me3 levels. **(C)** Inflammatory signals, especially CCR2 ligands, elevate CCR2 expression in myogenic progenitors, inhibiting their fusion into myofibers by activating MAPKp38$\delta$/$\gamma$ signaling and phosphorylating MyoD. Chronic inflammatory signals reduce Kmt5a activity, leading to a decline in H4K20me1 levels, disrupting MuSC quiescence by down-regulating Notch target genes. In addition, chronic inflammation disrupts p38$\alpha$/$\beta$ MAPK, adding repressive H3K27me3 marks on myogenic gene promoters such as Pax7.

### Neural stem cell (NSC)

NSCs play a pivotal role in neurogenesis, essential for maintaining brain plasticity throughout life. Neuroinflammation in the central nervous system is unique because of the blood-brain barrier, separating the CNS from the peripheral immune system when still allowing an immunological response. Microglia, the resident immune cells in the CNS, are pivotal in initiating innate immune responses through pattern recognition receptors that detect danger signals from dying cells and microorganisms. Upon activation, microglia undergo morphological changes and release nitric oxide, ROS, and various pro-inflammatory cytokines and chemokines, orchestrating a robust M1 response. This M1 response, can

lead to chronic inflammation if not properly regulated, with potential long-term consequences for surrounding neural cells (266). Conversely, M2 microglia release anti-inflammatory cytokines such as TGF-β and IL-10, promoting neuroprotection and tissue repair, although the M2 response is often transient in various CNS injury models.

LPS, commonly used to model inflammation in vitro, have been shown to adversely affect NSC viability and differentiation. In LPS-treated NSCs, key epigenetic changes occur, especially involving histone modifications. For instance, prolonged LPS exposure increases the expression of Jmjd2b, a histone demethylase, and NF-κB p65, indicating their roles in inflammation-induced epigenetic regulation (267) (Fig 5). Jmjd2b targets the trimethylated lysine on H3K9me3, a mark typically associated with repressed chromatin, and demethylates it to maintain an open chromatin state conducive to gene transcription. This epigenetic alteration was confirmed through expression analysis and ChIP assays, which revealed reduced H3K9me3 levels at promoters of genes such as *Notch1* and *IL-1β* in long-term LPS-treated cells. The knockdown of Jmjd2b led to decreased expression of various genes, including *p65*, *iNOS*, *Bcl2*, and *TGF-β*, highlighting its regulatory influence. Pathway analyses further identified down-regulated genes such as *NF-κB*, *Sema3e*, *Wnt*, and *Lef1*, involved in neurogenesis and cellular differentiation processes. Thus, inflammation through LPS-induced pathways profoundly alters the epigenetic landscape of NSCs, primarily via histone demethylation, which affects key genes and pathways essential for NSC function and neurogenesis. Understanding these mechanisms provides crucial insights into the interplay between inflammation and NSC regulation (268).

Alternatively, inflammation or injury can cause astrocytes to become reactive, reacquiring characteristics such as neural progenitor cells (NPCs), with pro-inflammatory cytokines such as TNF-α playing a key role in this reprogramming. Research exploring the transcriptome and epigenetic profiles of astrocyte populations has revealed that changes in epigenetic markers, such as histone modifications associated with active promoters, such as H3K4me3 (117), and with repressed promoters, such as H3K27me3 (118), accompany shifts in cellular potential (268). Experimental studies using the mouse NPC line CTX12 have demonstrated that astrocytes' plasticity and capacity for dedifferentiation are influenced by their differentiation conditions. This understanding of astrocytes' latent neurogenic potential and the epigenetic mechanisms regulating it is vital for developing regenerative medicine strategies. Notably, reactive astrocytes, naturally present at injury sites, offer a promising target for brain repair mechanisms, underscoring the role of inflammation in activating pathways that reprogram astrocytes to a more plastic, NPC-like state.

### MuSC
Aging-related systemic inflammation exerts profound effects on MuSCs, driving substantial alterations in their epigenetic landscape and contributing to their functional decline. Inflammatory signals, particularly those mediated through CCR2 ligands such as CCL2, CCL7, and CCL8, play a pivotal role in this process. These signals elevate the levels of CCR2 chemokines, leading to increased CCr2 expression in myogenic progenitors. This heightened CCR2 activity inhibits MP fusion into myofibers by activating

MAPKp38δ/γ signaling and phosphorylating MyoD, which suppresses myogenic commitment factor Myogenin. As a result, the regenerative capacity of muscle is impaired, as myogenic progenitors are less able to form the multinucleated myotubes needed for effective muscle repair exacerbating the decline in muscle function seen with aging (269).

Chronic exposure to these inflammatory signals leads to a significant reduction in the activity of Kmt5a, an enzyme crucial for the monomethylation H4K20. The diminished activity of Kmt5a and the resultant decline in H4K20me1 disrupt MuSC quiescence by down-regulating Notch target genes, essential for maintaining stem cell dormancy. This disruption is exacerbated in aged MuSCs, which exhibit lower levels of both Kmt5a and H4K20me1, creating an epigenetic environment that predisposes them to premature activation and subsequent cell death via ferroptosis (270 Preprint). Reduced Kmt5a activity leads to decreased levels of H4K20me1, a histone modification critical for the formation of constitutive heterochromatin and proper cell-cycle progression. This reduction disrupts the transcriptional programs necessary for MuSC quiescence and survival. Specifically, the loss of H4K20me1 results in the repression of Notch signaling components, including Notch, Jag1, Numb, and Rbpj, through increased promoter-proximal pausing, leading to decreased expression of quiescence-associated genes, such as *Hes1* and *Hey1*, and increased expression of activation-associated genes, such as *Myf5* and *Dek*. Furthermore, the persistent reduction in H4K20me1 prime MuSCs for ferroptosis, characterized by abnormal iron metabolism, elevated intracellular iron levels, increased ROS, and lipid peroxidation. This epigenetic reprogramming, driven by inflammation, sensitizes MuSCs to inflammatory signals, leading to their premature exit from quiescence, susceptibility to ferroptosis, and impaired regenerative capacity.

In young muscle, injury induces a coordinated inflammatory response, starting with the recruitment of neutrophils, eosinophils, and macrophages. This triggers a pro-inflammatory phase that promotes myogenic proliferation through cytokines such as IL-6 and TNF-α (113, 271) followed by an anti-inflammatory phase aiding differentiation and repair via TGF-β and IGF-1 (272, 273). However, in aged muscle this becomes chronic, leading to persistent inflammation that hampers regeneration (274) (Fig 6C). For example, increased IL-6 signaling in aged mice causes MuSC exhaustion and atrophy. Epigenetically, chronic inflammation disrupts p38α/β MAPK, linked to PRC2, adding repressive H3K27me3 marks on myogenic gene promoters such as *Pax7* (247). After activation by TNF-α, the p38α kinase phosphorylates EZH2 which enhances the binding affinity of EZH2 to transcription factor YY1. The p38α, YY1, and EZH2 complex is recruited to the *Pax7* promoter, catalyzing the trimethylation of H3K27. PAX7 directly influences myogenic gene expression by inducing chromatin modifications through histone methyltransferase complexes (275). Notably, PAX7 up-regulates *Myf5*, a key regulator of muscle commitment, via association with the Wdr5-Ash2L-MLL2 HMT complex, which methylates histone H3K4. This marks chromatin as active, facilitating the activation of MYF5. ChIP studies confirmed that PAX7 recruits this complex to gene promoters, promoting H3K4 methylation and ensures transcription of myogenic genes. In addition, PAX7's association with demethylated and trimethylated H3K4 regions shows its role in maintaining active chromatin states.

In pathological conditions such as muscular dystrophies, persistent inflammation drives extensive epigenetic reprogramming. Chronic inflammation in these diseases elevates levels of cytokines that continuously activate pathways leading to both DNA methylation and histone modifications. This reprogramming not only suppresses myogenic differentiation but also enhances fibrogenic pathways, shifting the balance towards fibrosis and muscle degeneration (276).

# Conclusion

Adult stem cells experience major forms of cellular stress such as replicative, oxidative, mechanical, and inflammatory stress, and these stresses have been reported to mostly elicit significant negative impacts detrimental to stem cell function via epigenetic alterations. However, depending on the stress type, namely mechanical stress, it is not especially harmful and, rather, may regulate certain stem cell populations and promote regenerative effects. Replicative stress, resulting from repeated cell division, induces epigenetic alterations on the DNA and, largely, histone levels in HSC, MSC, and ISCs. Histone modifiers Ott1, PcG proteins (i.e., EZH2), and CBP in HSCs mitigate replicative stress, preventing phenotypes such as increased expansion, skewed differentiation bias, and increased susceptibility to cell death and senescence. Histone variant substitution was found to only take place in HSC and ISCs involving DDR markers γH2AX and macroH2A1.1, respectively, under replicative stress outside of other cellular stressors and adult stem cells in literature. Both substituted variants accumulate and are suggested to be established permanently, leading to the impaired stem cell maintenance and regenerative capacities. Oxidative stress, caused by the imbalance between ROS and antioxidant defense mechanisms, can damage cellular macromolecules and reduce stem cell viability and differentiation capacity. Mechanical stress, arising from physical forces acting on stem cells, can alter cell shape, cytoskeletal organization, and gene expression, affecting stem cell fate decisions. A few of these stress-induced phenotypes are also seen in a similar fashion in an aging context. These include a skewed differentiation output towards the myeloid lineage and increased expansion under oxidative stress in young HSCs and dysregulation of organelle homeostasis in MSCs under replicative stress. Oxidative stress can also cause epigenetic alterations, including DNA damage and modification of histone proteins, resulting in changes in chromatin structure and gene expression. Furthermore, mechanical stress can affect epigenetic regulation by altering the chromatin conformation, DNA accessibility, and gene expression profiles. Studies have shown that these stresses can lead to changes in the expression of key epigenetic regulators, including DNMTs and histone-modifying enzymes, which in turn can influence stem cell differentiation and self-renewal. Inflammation can also exert significant epigenetic effects on al stem cells, such as altering DNA methylation patterns and histone modifications, which may impair their regenerative capacity and promote senescence. Chronic inflammatory signals can lead to persistent epigenetic changes that disrupt stem cell function and contribute to age-related decline in tissue regeneration. The reported stresses discussed can occur uniquely or in combination, leading to complex interactions and effects on adult stem cells. Understanding the consequences of stress has important implications for optimizing or developing novel therapeutic strategies and protocols, such as stem cell transplantation therapies and expansion protocols, as there may be negative consequences of enforcing robust expansion of stem cells. Thus, understanding how these stresses impact epigenetic regulation in adult stem cells is critical for developing strategies to optimize their regenerative potential in tissue repair and regeneration. To visually summarize the epigenetic changes in each cell type, we created detailed figures for HSCs (Fig 2), MSCs (Fig 3), ISCs (Fig 4), NSCs (Fig 5), and MuSCs (Fig 6). Further research is needed to elucidate the precise mechanisms by which these stresses affect epigenetic regulation in stem cells and to identify potential therapeutic interventions to minimize their negative effects on stem cell function.

# Supplementary Information

# Acknowledgements

This manuscript was supported by Intramural Research Funds from the NIA/NIH.

## Author Contributions

J Llewellyn: conceptualization and writing—original draft, review, and editing.
R Baratam: conceptualization, supervision, validation, and writing—original draft, review, and editing.
L Culig: conceptualization, resources, supervision, funding acquisition, validation, project administration, and writing—original draft, review, and editing.
I Beerman: conceptualization, resources, supervision, funding acquisition, project administration, and writing—original draft, review, and editing.

## Conflict of Interest Statement

The authors declare that they have no conflict of interest.

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
