## [Reviewer comments · Life Science Alliance]

Life Science Alliance

Cellular stress and epigenetic regulation in adult stem cells

Joey Llewellyn, Rithvik Baratam, Luka Culig, and Isabel Beerman

DOI: <https://doi.org/10.26508/lsa.202302083>

Corresponding author(s): Isabel Beerman, National Institute on Aging

Review Timeline:

Submission Date:	2023-04-07
Editorial Decision:	2023-05-30
Revision Received:	2024-07-25
Editorial Decision:	2024-07-26
Revision Received:	2024-09-16
Accepted:	2024-09-16

Transaction Report:

May 30, 2023

Re: Life Science Alliance manuscript #LSA-2023-02083

Dr. Isabel Beerman
National Institute on Aging
251 Bayview Blvd
Baltimore, MD 21224

Dear Isabel,

Thank you for submitting your manuscript entitled "Cellular stress and epigenetic regulation in adult stem cells" to Life Science Alliance. The manuscript was assessed by expert reviewers, whose comments are appended to this letter. As you can notice all reviewers are quite positive and see this review as very timely. They highlight few concerns that can be easily fixed, so overall very constructive. We therefore invite you to submit a revised manuscript addressing the Reviewer comments.

Thank you for this interesting contribution to Life Science Alliance. We are looking forward to receiving your revised manuscript.

Sincerely,

B. MANUSCRIPT ORGANIZATION AND FORMATTING:

Reviewer #1 (Comments to the Authors (Required)):

Beerman and colleagues have written a very timely and interesting review on stress inducers and epigenetic modification in adult stem cell. The topic is very timely. There is growing evidence that stem cell aging (the decline in stem cell function during lifetime) is strongly influenced by stress. How stress factors that occur during lifetime impact on the functional decline of stem cells is not well understood. It is anticipated that epigenetic alterations in response to stress could play an important role in this process. Beerman and colleagues nicely focus on this main theme the role of stress induced epigenetic alteration in stem cells. They mainly focus on hematopoietic, neural, muscle and intestinal stem cells as well as on mesenchymal stromal cells - a not well defined cell population that includes stem cells. I only have a few comments that the authors may want to address:

1. Line 56-60: "Cellular stress affects the cell body, proteomic, and DNA levels, and is elicited stressors originating extra- or intracellularly." I am not sure what the authors mean with DNA levels, the genomic DNA should stay the same (in quantity), do they mean modifications or mitochondrial DNA?
2. Lines 122-126: The authors discuss differences in the functional role of surface markers on stem cells. Would it be interesting to include a short discussion on CD34 in human vs. mouse HSC?
3. Line 222 - the authors mention a pre-print. Is this paper still a pre-print? Is it important for the review? I find it a bit misplaced.
4. Lines 237-239: The authors describe: "ISCs continuously undergo rapid asymmetrical cell division in the base of crypts to give rise to daughter cells known as transit amplifying (TA) cells." I am not sure about this statement. There is literature indicating that ISCs compete for space in basal crypt niches, using symmetric cell division, see for example: "Intestinal Crypt Homeostasis Results from Neutral Competition between Symmetrically Dividing Lgr5 Stem Cells" from the Clevers lab and "Intestinal crypt homeostasis revealed at single-stem-cell level by in vivo live imaging".
5. Line 296-297 on muscle stem cells: "In addition to growth, these stem cells are also essential for muscle maintenance. This is a bit under debate. There is literature on MuSC depletion in genetic mouse models, leading to no obvious defects in muscle maintenance. See for example: "Effective fiber hypertrophy in satellite cell-depleted skeletal muscle" by McCarthy et al. 2011
6. An additional topic that could be covered in the section replication stress, is the transit from quiescence to cell cycle, for example in MuSC and HSC. There is evidence that this transitional stage by itself can be misregulated during aging, which seems to associate with alteration in epigenetic modification, see for example "Epigenetic stress responses induce muscle stem-cell ageing by Hoxa9 developmental signals".
7. Another topic of interest, is the evidence of early life stress during development or in response to inflammatory response, that leads to memory effects in stem cells influencing stem cell aging, see for example "Age-dependent effects of Igf2bp2 on gene regulation, function, and aging of hematopoietic stem cells in mice", "Inflammatory exposure drives long-lived impairment of hematopoietic stem cell self-renewal activity and accelerated aging" or "C/EBP β -dependent epigenetic memory induces trained immunity in hematopoietic stem cells"

Reviewer #2 (Comments to the Authors (Required)):

In this review, Llewellyn et al. explore the effects of various stresses on different stem cell types, the epigenetic mechanisms involved in the cellular responses to these perturbations and how these could be applied to therapeutics and future research. This is a timely topic for review and could be of interest to a broad readership given the recent interest in how cellular stress (infection/inflammation, malignancies, oxidative etc.) can impact cells at the epigenetic level and what this means for the organism in the long-term.

My main comment is that, while this piece is very well written and extremely comprehensive, it felt more like I was reading the introduction to a thesis. There is a lot of information here and it was unclear to me what the point of view of this review is. In its current form, it does not highlight the most recent or novel developments in this area. This is not to discredit the effort and work that has gone into writing this piece, I just feel that it could be more concise and streamlined and needs a clearer direction/story to appeal to maximum readers. I questioned the relevance of certain parts and there was a lot of repetition in areas. I was also expecting some mention about infection/inflammation/disease as a type of stress and some mention of how this impacts cells at the epigenetic level. The conclusions feel very broad and general. Also, the table is useful but would be better to include references if possible.

There definitely is potential here for a great review, it just needs some work to make it a more concise synopsis of the current state of the field and future outlook to make it the best possible resource for the readers.

Reviewer #3 (Comments to the Authors (Required)):

The review by Llewellyn et al. describes the recent insights in stem cells biology and stress.. This group has digested an impressive literature base and summarized those findings in this comprehensive review. The submission overall was well thought out, but a few suggestions are being made to improve the overall flow and storytelling. Overall, this was a pleasure to read.

Minor revisions/suggestions:

1. Most figures are not related to the main topic, and although they were well done, some minor modifications to link to the text

would be encouraged. Overall, these figures don't visually show the stress related events in stem cells, or mechanisms to overcome this.

2. The stem cell sections were very detailed and suitable in describing different stem cell types, as well as parts describing the stress and stress responses; however, many of the "stress" sections were under-explain certain points (perhaps due to a lack of literature in this area, so maybe just say that).

3. One comment on the structure, the background section on "Epigenetic Mechanisms" should be eliminated and simply incorporated into the specific stem cell sections that address those topics. Currently, they are written as an overview, but not of stem cells or stress, just background, so with little effort this could be corrected.

Reviewer #1 (Comments to the Authors (Required)):

Beerman and colleagues have written a very timely and interesting review on stress inducers and epigenetic modification in adult stem cell. The topic is very timely. There is growing evidence that stem cell aging (the decline in stem cell function during lifetime) is strongly influenced by stress. How stress factors that occur during lifetime impact on the functional decline of stem cells is not well understood. It is anticipated that epigenetic alterations in response to stress could play an important role in this process. Beerman and colleagues nicely focus on this main theme the role of stress induced epigenetic alteration in stem cells. They mainly focus on hematopoietic, neural, muscle and intestinal stem cells as well as on mesenchymal stromal cells - a not well-defined cell population that includes stem cells. I only have a few comments that the authors may want to address:

Thank you to the reviewer for these kind comments and suggestions. We have attempted to incorporate the suggestions in the revised review: please see direct comments below.

1. Line 56-60: "Cellular stress affects the cell body, proteomic, and DNA levels, and is elicited stressors originating extra- or intracellularly. " I am not sure what the authors mean with DNA levels, the genomic DNA should stay the same (in quantity), do they mean modifications or mitochondrial DNA?

We have edited the statement to read, "Cellular stress affects the cell body, proteome, genome, and epigenome, and is elicited by stressors originating extra- or intracellularly."

2. Lines 122-126: The authors discuss differences in the functional role of surface markers on stem cells. Would it be interesting to include a short discussion on CD34 in human vs. mouse HSC?

We now include a few sentences about the differences in CD34 expression, (lines 138-142) "One of the most striking differences between human and mouse immunophenotypic markers is CD34. In the murine system, HSCs are classified as CD34⁻ while human HSCs are characterized as CD34⁺. Intriguingly, a population of CD34⁻ human HSCs has recently been identified in cord blood (DOI: 10.1038/s41467-018-04441-z) however it is unclear if this population of megakaryocyte/ erythrocyte biased HSCs also exists in adult bone marrow." in addition to the discussion of Flt-3.

3. Line 222 - the authors mention a pre-print. Is this paper still a pre-print? Is it important for the review? I find it a bit misplaced.

The manuscript is now published in Nature Aging, but we have removed this comment, as it was not entirely relevant to this review.

4. Lines 237-239: The authors describe: "ISCs continuously undergo rapid asymmetrical cell division in the base of crypts to give rise to daughter cells known as transit amplifying (TA) cells." I am not sure about this statement. There is literature indicating that ISCs compete for space in basal crypt niches, using symmetric cell division, see for example: "Intestinal Crypt

Homeostasis Results from Neutral Competition between Symmetrically Dividing Lgr5 Stem Cells" from the Clevers lab and "Intestinal crypt homeostasis revealed at single-stem-cell level by in vivo live imaging".

Thank you for pointing this out- We have amended this statement to more accurately reflect the literature in lines 253-263.

5. Line 296-297 on muscle stem cells: "In addition to growth, these stem cells are also essential for muscle maintenance. This is a bit under debate. There is literature on MuSC depletion in genetic mouse models, leading to no obvious defects in muscle maintenance. See for example: "Effective fiber hypertrophy in satellite cell-depleted skeletal muscle" by McCarthy et al. 2011

We have edited this section to remove the "essential to maintenance" statement.

6. An additional topic that could be covered in the section replication stress, is the transit from quiescence to cell cycle, for example in MuSC and HSC. There is evidence that this transitional stage by itself can be misregulated during aging, which seems to associate with alteration in epigenetic modification, see for example "Epigenetic stress responses induce muscle stem-cell ageing by Hoxa9 developmental signals".

We have now included a short section discussing this transition in the replication stress section.

7. Another topic of interest, is the evidence of early life stress during development or in response to inflammatory response, that leads to memory effects in stem cells influencing stem cell aging, see for example "Age-dependent effects of Igf2bp2 on gene regulation, function, and aging of hematopoietic stem cells in mice", "Inflammatory exposure drives long-lived impairment of hematopoietic stem cell self-renewal activity and accelerated aging" or "C/EBP β -dependent epigenetic memory induces trained immunity in hematopoietic stem cells"

We have incorporated a brief section highlighting the evidence of early life stress/inflammatory stimuli inducing memory effects in stem cells and subsequently influence stem cell aging in our discussion of the effects of inflammatory stress on HSCs (lines 946-966).

Reviewer #2 (Comments to the Authors (Required)):

In this review, Llewellyn et al. explore the effects of various stresses on different stem cell types, the epigenetic mechanisms involved in the cellular responses to these perturbations and how these could be applied to therapeutics and future research. This is a timely topic for review and could be of interest to a broad readership given the recent interest in how cellular stress (infection/inflammation, malignancies, oxidative etc.) can impact cells at the epigenetic level and what this means for the organism in the long-term.

My main comment is that, while this piece is very well written and extremely comprehensive, it felt more like I was reading the introduction to a thesis. There is a lot of information here and it was unclear to me what the point of view of this review is. In its current form, it does not

highlight the most recent or novel developments in this area. This is not to discredit the effort and work that has gone into writing this piece, I just feel that it could be more concise and streamlined and needs a clearer direction/story to appeal to maximum readers. I questioned the relevance of certain parts and there was a lot of repetition in areas. I was also expecting some mention about infection/inflammation/disease as a type of stress and some mention of how this impacts cells at the epigenetic level. The conclusions feel very broad and general. Also, the table is useful but would be better to include references if possible.

There definitely is potential here for a great review, it just needs some work to make it a more concise synopsis of the current state of the field and future outlook to make it the best possible resource for the readers.

Thank you for your thorough review of our manuscript. We appreciate your constructive feedback, which has significantly contributed to improving the quality of our review. In response to your comments, we have made several revisions to enhance the focus and clarity of our manuscript. We have deleted a background section on epigenetics and included a summary introduction to streamline the discussion of specific epigenetic changes. We have also addressed the issue of repetition and refined the content to provide a more concise and directed narrative. In addition, we have added a dedicated section on infection, inflammation, and disease as types of stress and their impact on cells at the epigenetic level. References have been incorporated into the table to improve its utility. We believe these changes address your concerns and have strengthened the manuscript, making it a more valuable resource for readers. We are confident that the revised version now provides a clearer and more focused synopsis of the current state of the field.

Reviewer #3 (Comments to the Authors (Required)):

The review by Llewellyn et al. describes the recent insights in stem cells biology and stress.. This group was digest an impressive literature base and summarize those findings in this comprehensive review. The submission overall was well thought out, but a few suggestions are being made to improve the overall flow and storytelling. Overall, this was a pleasure to read.

Minor revisions/suggestions:

1. Most figures are not related to the main topic, and although they were well done, some minor modifications to link to the text would be encouraged. Overall, these figures don't visually show the stress related events in stem cells, or mechanisms to overcome this.

We have done our best to revise figures to make them more useful to the readers and appreciate this suggestion.

2. The stem cell sections were very detailed and suitable in describing different stem cell types, as well as parts describing the stress and stress responses; however, many of the "stress" sections were under-explain certain points (perhaps due to a lack of literature in this area, so maybe just say that).

We appreciate this feedback and have aimed to highlight the complexities of stress responses in stem cells. However, the current literature does lack comprehensive studies in certain areas, and gaps in literature base have proved challenging. We are hoping to point out that these are key missing analyses and use this review to also highlight the importance of future studies evaluating stress on stem cells.

3. One comment on the structure, the background section on "Epigenetic Mechanisms" should be eliminated and simply incorporated into the specific stem cell sections that address those topics. Currently, they are written as an overview, but not of stem cells or stress, just background, so with little effort this could be corrected.

Thank you for the suggestion. We have removed the standalone background section on 'Epigenetic Mechanisms' and incorporated a summary into the relevant stem cell sections to provide better context and coherence.

July 26, 2024

RE: Life Science Alliance Manuscript #LSA-2023-02083R

Dr. Isabel Beerman
National Institute on Aging
251 Bayview Blvd
Baltimore, MD 21224

Dear Dr. Beerman,

Thank you for submitting your revised manuscript entitled "Cellular stress and epigenetic regulation in adult stem cells". We would be happy to publish your paper in Life Science Alliance pending final revisions necessary to meet our formatting guidelines.

- please be sure that the authorship listing and order is correct
- please add the Twitter handle of your host institute/organization as well as your own or/and one of the authors in our system
- please add an Author Contributions section to your main manuscript text
- please add a Conflict of Interest statement to your main manuscript text
- please add your figure legends to the main manuscript text after the references section
- please add callouts for Figures 2A-C; 3A-C; 4A-C and 6A-C to your main manuscript text

A. FINAL FILES:

B. MANUSCRIPT ORGANIZATION AND FORMATTING:

**Submission of a paper that does not conform to Life Science Alliance guidelines will delay the acceptance of your

manuscript.**

The license to publish form must be signed before your manuscript can be sent to production. A link to the electronic license to publish form will be available to the corresponding author only. Please take a moment to check your funder requirements.

Sincerely,

September 16, 2024

RE: Life Science Alliance Manuscript #LSA-2023-02083RR

Dr. Isabel Beerman
National Institute on Aging
251 Bayview Blvd
Baltimore, MD 21224

Dear Dr. Beerman,

Thank you for submitting your Commissioned Review entitled "Cellular stress and epigenetic regulation in adult stem cells". It is a pleasure to let you know that your manuscript is now accepted for publication in Life Science Alliance. Congratulations on this interesting work.

Again, congratulations on a very nice paper. I hope you found the review process to be constructive and are pleased with how the manuscript was handled editorially. We look forward to future exciting submissions from your lab.

Sincerely,
